# FoxO3 an important player in fibrogenesis and therapeutic target for idiopathic pulmonary fibrosis

Hamza M Al-Tamari[1] ⬤, Swati Dabral[1,†], Anja Schmall[1,†], Pouya Sarvari[1], Clemens Ruppert[2], Jihye Paik[3], Ronald A DePinho[4], Friedrich Grimminger[2], Oliver Eickelberg[5], Andreas Guenther[2,6], Werner Seeger[1,2], Rajkumar Savai[1,2] & Soni S Pullamsetti[1,2,*] ⬤

## Abstract

Idiopathic pulmonary fibrosis (IPF) is a progressive and fatal parenchymal lung disease with limited therapeutic options, with fibroblast-to-myofibroblast transdifferentiation and hyperproliferation playing a major role. Investigating *ex vivo*-cultured (myo)fibroblasts from human IPF lungs as well as fibroblasts isolated from bleomycin-challenged mice, Forkhead box O3 (FoxO3) transcription factor was found to be less expressed, hyperphosphorylated, and nuclear-excluded relative to non-diseased controls. Downregulation and/or hyperphosphorylation of FoxO3 was reproduced by exposure of normal human lung fibroblasts to various pro-fibrotic growth factors and cytokines (FCS, PDGF, IGF1, TGF-β1). Moreover, selective knockdown of FoxO3 in the normal human lung fibroblasts reproduced the transdifferentiation and hyperproliferation phenotype. Importantly, mice with global-(*Foxo3*[−/−]) or fibroblast-specific (*Foxo3*[f.b][−/−]) FoxO3 knockout displayed enhanced susceptibility to bleomycin challenge, with augmented fibrosis, loss of lung function, and increased mortality. Activation of FoxO3 with UCN-01, a staurosporine derivative currently investigated in clinical cancer trials, reverted the IPF myofibroblast phenotype *in vitro* and blocked the bleomycin-induced lung fibrosis *in vivo*. These studies implicate FoxO3 as a critical integrator of pro-fibrotic signaling in lung fibrosis and pharmacological reconstitution of FoxO3 as a novel treatment strategy.

**Keywords** fibroblast; forkhead box O transcription factors; idiopathic pulmonary fibrosis; myofibroblast; transdifferentiation

**Subject Categories** Pharmacology & Drug Discovery; Respiratory System

## Introduction

Idiopathic pulmonary fibrosis (IPF) is a lethal, progressive fibrosing parenchymal lung disease, which affects millions of patients worldwide and is largely refractory to current treatment. The pathogenesis of IPF is not well understood and continues to be explored. IPF is characterized by the presence of hyperproliferative fibroblasts/myofibroblasts and increased deposition of extracellular matrix (ECM), thereby distorting normal lung architecture and causing loss of respiratory function (Selman & Pardo, 2002).

Fibroblasts are the most versatile of the connective tissue cell family and possess a remarkable capacity to undergo various phenotypic conversions between distinct, but related cell types (White *et al*, 2003). In addition, fibroblasts represent the key effector cells in wound healing, where they secrete ECM proteins that provide a tissue scaffold for normal repair events. Eventual dissolution of this scaffold and apoptosis of fibroblasts/myofibroblasts are critical for restoration of normal tissue architecture (Lorena *et al*, 2002; White *et al*, 2003). In IPF, however, fibroblasts are phenotypically different, presenting a highly contractile and synthetic profile (myofibroblasts), linked with aberrant wound healing and disturbed epithelial–mesenchymal cross talk. Numerous reports have demonstrated that myofibroblasts isolated from IPF patients, when compared to normal fibroblasts, display several morphological and functional abnormalities, including a spindle or stellate morphology with α-smooth muscle actin (α-SMA) stress fibers, hyperproliferative potential, excessive synthesis and remodeling of extracellular matrix, and expression of growth factors and cytokines that drive fibrogenesis. Moreover, myofibroblast activation and tissue remodeling are an ongoing event in IPF, in contrast to normal wound healing. Hence, myofibroblasts are considered as primary effector cell for matrix deposition and tissue remodeling in IPF (Moodley *et al*, 2003; Xia *et al*, 2008). However, the molecular mechanisms underlying the IPF (myo)fibroblast phenotype have only recently begun

---

1 Department of Lung Development and Remodeling, Max-Planck-Institute for Heart and Lung Research, Member of the German Center for Lung Research (DZL), Bad Nauheim, Germany
2 Department of Internal Medicine, Universities of Giessen and Marburg Lung Center (UGMLC), Member of the DZL, Justus-Liebig University, Giessen, Germany
3 Department of Pathology and Laboratory medicine, Weill Cornell Medical College, New York City, NY, USA
4 Division of Basic Science Research, Department of Cancer Biology, The University of Texas MD Anderson Cancer Center, Houston, TX, USA
5 Comprehensive Pneumology Center, Ludwig Maximilians University Munich and Helmholtz Zentrum München, Munich, Germany
6 AGAPLESION Lung Clinic Waldhof-Elgershausen, Greifenstein, Germany
 *Corresponding author. Tel: +49 6032 705 380; Fax: +49 6032 705 385; E-mail: soni.pullamsetti@mpi-bn.mpg.de
 †These authors contributed equally to this work

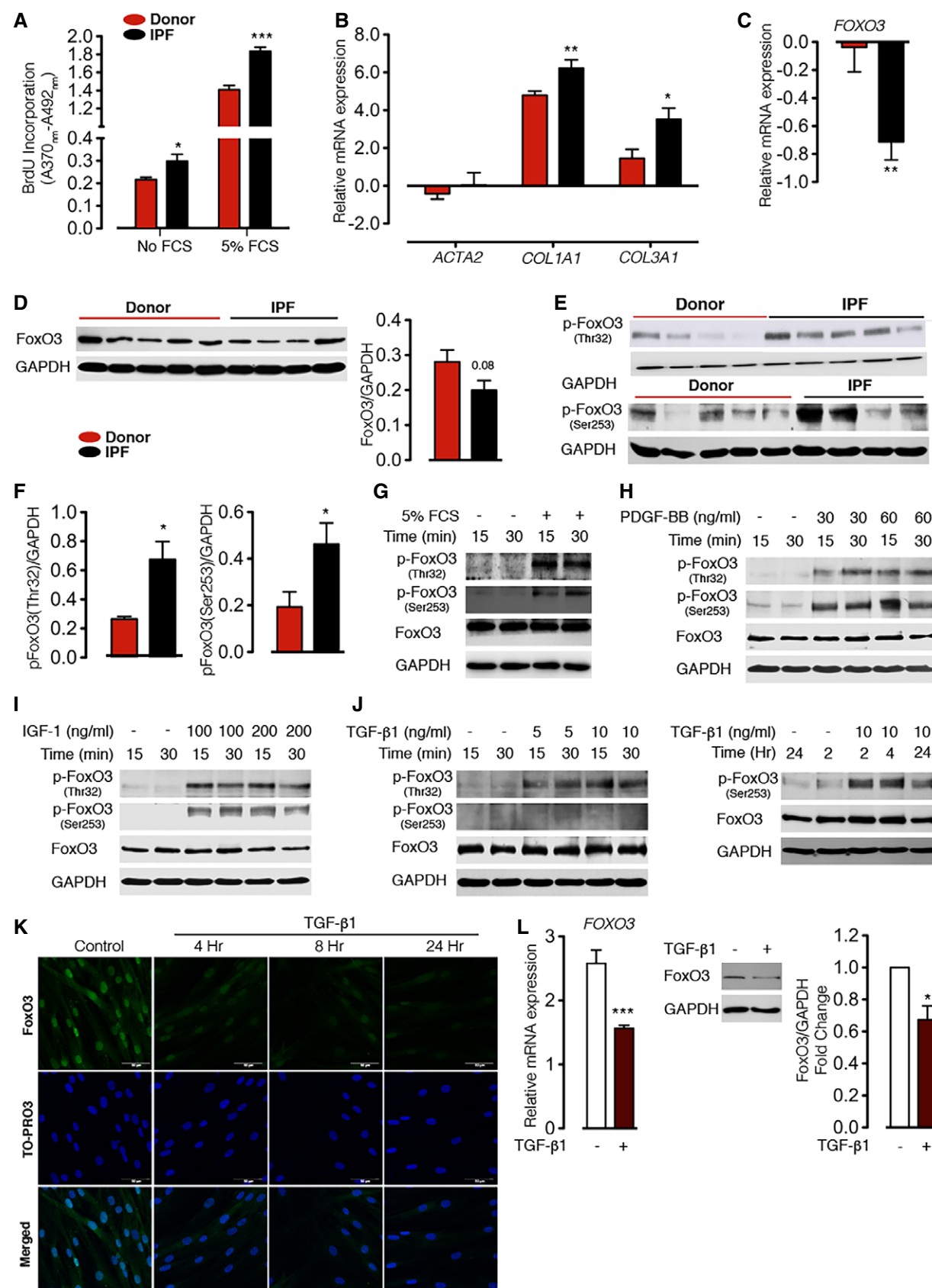

**Figure 1.**

to be elucidated. Elevated levels of several pro-fibrotic growth factors and cytokines [such as platelet-derived growth factor (PDGF), insulin-like growth factor 1 (IGF-1), and transforming growth factor-β1 (TGF-β1)] have been reported to promote this pathological phenotype (Uh *et al*, 1998; Bonner, 2004; Gauldie *et al*, 2007).

FoxOs belong to a family of transcriptional regulators characterized by a conserved DNA-binding domain termed the forkhead box (Lam *et al*, 2013). FoxOs, when present in the nucleus and bound to promoters that contain the FoxO consensus motif, can act as transcriptional activators and repressors. In mammals, four FoxO isoforms have been identified: FoxO1, FoxO3, FoxO4, and FoxO6 (Lam *et al*, 2013). Among these, FoxO3 has been shown to play an essential role in various biological processes, including development, proliferation, apoptosis, metabolism, and differentiation, by regulating a wide spectrum of genes. The FoxOs are functional redundant tumor suppressors (Paik *et al*, 2007) and are genomically or functionally inactivated in diverse cancer types, including lung cancer (Brunet *et al*, 1999; Hu *et al*, 2004).

Here, we show that FoxO3 plays a crucial role in IPF fibrogenesis. FoxO3 was found to be downregulated in IPF (myo)fibroblasts, and this was reproduced by *ex vivo* exposure of normal human lung fibroblasts to various pro-fibrotic growth factors and cytokines via activation of phosphoinositide 3-kinase (PI3K)/Akt signaling. Moreover, shRNA-mediated knockdown of FoxO3 in normal human lung fibroblasts provoked their transdifferentiation to hyperproliferative myofibroblasts. Correspondingly, global- ($Foxo3^{-/-}$) and fibroblast-specific ($Foxo3_{f.b}^{-/-}$) FoxO3 knockout mice exhibited enhanced susceptibility to bleomycin challenge, with augmented fibrosis, loss of lung function, and increased mortality. Partial pharmacological restoration of FoxO3 activity can attenuate the IPF myofibroblast phenotype *in vitro* and bleomycin-induced lung fibrosis *in vivo*, illuminating a potential therapeutic strategy.

## Results

### FoxO3 is downregulated and hyperphosphorylated in IPF fibroblasts

To study FoxO3 regulation in IPF, we isolated primary human lung fibroblast from healthy donors (hereafter referred to as N-HLF) and patients with IPF (hereafter referred to as IPF-HLF). Previous reports have shown that IPF fibroblasts exhibit a highly proliferative phenotype compared to donor fibroblasts. Additionally, they express high levels of myofibroblast markers such as collagen 1a1 (Col1a1), collagen 3a1 (Col3a1), and α-SMA (Ramos *et al*, 2001). Consistent with these reports, we observed an increase in the proliferative capacity of IPF-HLF compared to N-HLF under basal and growth factor-stimulated conditions (Fig 1A). Additionally, IPF-HLF displayed a significantly increased expression of the *COL1A1* and *COL3A1* genes, with no changes in the expression levels of *ACTA2* (encoded α-SMA; Fig 1B). Interestingly, when we analyzed FoxOs mRNA expression, *FOXO3* was the most abundant form (Appendix Fig S1A) and was significantly downregulated in IPF-HLF compared to N-HLF (Fig 1C), whereas the level of *FOXO1* and *FOXO4* were unchanged (Appendix Fig S1B). Similarly, the protein level of FoxO3, although not significantly, was substantially downregulated in IPF-HLF (Fig 1D). Changes of FoxO3 activity were assessed by measuring the phosphorylation status of $Thr^{32}$ and $Ser^{253}$ of FoxO3, sites that are phosphorylated by the PI3K/Akt pathway resulting in the inactivation of FoxO3 (Brunet *et al*, 1999). The levels of phosphorylated FoxO3 (p-FoxO3) at $Thr^{32}$ and $Ser^{253}$ were significantly higher in IPF-HLF, as compared to N-HLF (Fig 1E and F).

### Growth factors and cytokines drive FoxO3 downregulation in human lung fibroblasts

Next, we assessed the effect of elevated growth factors in IPF such as PDGF-BB and IGF-1 on FoxO3 activity in human lung fibroblasts (Bonner, 2004; Hetzel *et al*, 2005). Interestingly, FoxO3 phosphorylation at $Thr^{32}$ and $Ser^{253}$ significantly increased in N-HLF in response to FCS, PDGF-BB, and IGF-I in a time- and dose-dependent manner (Figs 1G–I and EV1A, Appendix Figs S2A and S3A). Moreover, FCS, PDGF-BB, and IGF-I stimulation resulted in FoxO3 nuclear exclusion in a time-dependent manner, with a complete exclusion after 6 h of stimulation (Fig EV1B, Appendix Figs S2B and S3B). All three stimulations also induced FoxO3 phosphorylation to a similar extent in IPF-HLFs (Appendix Fig S4) and N-HLFs (Appendix Fig S5). Interestingly, induction of FoxO3 phosphorylation was independent of N-HLF line (Fig EV2) as well their passage (Appendix Fig S6).

TGF-β1 is another elevated fibrotic factor in IPF that plays an important role in IPF pathogenesis by inducing fibroblast-to-

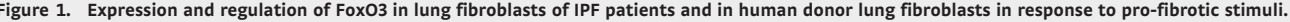

**Figure 1. Expression and regulation of FoxO3 in lung fibroblasts of IPF patients and in human donor lung fibroblasts in response to pro-fibrotic stimuli.**

A    Proliferation rate of N-HLF (isolated from healthy donor; *n* = 3) and IPF-HLF (isolated from IPF patients; *n* = 3) in the absence or presence of 5% FCS was measured by BrdU incorporation.

B    mRNA expression of myofibroblasts markers (*ACTA2, COL1A1,* and *COL3A1*) in N-HLF (*n* = 9) and IPF-HLF (*n* = 7) by qPCR.

C–F    Expression and activity of FoxO3 in N-HLF and IPF-HLF (passages 4–5). (C) mRNA expression analysis of *FOXO3* by qPCR (*n* = 7/group). (D, left panel) Representative Western blot of FoxO3. (D, right panel) Densitometrically quantified data of FoxO3/GAPDH expression ratio (*n* = 6–7/group). (E) Representative Western blots of p-FoxO3 (Thr32), p-FoxO3 (Ser253). (F) Densitometrically quantified data of p-FoxO3 (Thr32) or p-FoxO3 (Ser253) to GAPDH expression ratio (*n* = 6–7/group).

G–J    Representative Western blots of p-FoxO3 (Thr32), p-FoxO3 (Ser253), and FoxO3 in serum-starved (48 h) N-HLF (*n* = 3) that were stimulated with 5% FCS (G), PDGF-BB (H), IGF-1 (I), or TGF-β1 (J) as indicated.

K    Representative immunocytochemistry (ICC) images of FoxO3 localization in N-HLF (*n* = 4) stimulated without/with TGF-β1 (10 ng/ml) as indicated. Scale bar = 50 μm.

L    mRNA expression of *FOXO3* by qPCR (*n* = 7; left panel) and (middle panel) representative Western blot of FoxO3 in N-HLF stimulated with TGF-β1 for 24 h (right panel). Densitometrically quantified data, represented as fold change of FoxO3/GAPDH expression ratio (*n* = 4).

Data information: Data are expressed as mean ± SEM. In (A–D and F), data were analyzed using Student's *t*-test, *P* < 0.05, **P* < 0.01, ***P* < 0.001 versus donor. In (L), data were analyzed using paired *t*-test, *P* < 0.05, ***P* < 0.001 versus control. In all blots, GAPDH was used as a loading control.

myofibroblast differentiation (Wang *et al*, 2006). Stimulation with TGF-β1 for 15 and 30 min in N-HLF strongly increased FoxO3 phosphorylation at the Thr$^{32}$ site, whereas phosphorylation at the Ser$^{253}$ site remains unchanged (Fig 1J; Appendix Fig S7). On the other hand, when stimulated with TGF-β1 for longer time period (2, 4, and 24 h), phosphorylation at Ser$^{253}$ site also increased (Fig 1J). Along the same line, increased phosphorylation of AKT at Thr$^{308}$ was also observed (Appendix Fig S8). Interestingly, longer time stimulations with TGF-β1 also resulted in decreased FoxO3 expression and immunoreactivity, however, with no clear FoxO3 nuclear exclusion (Fig 1K and L).

### FoxO3 knockdown in healthy donor fibroblasts *in vitro* reproduces IPF fibroblast phenotype

To examine the effect of FoxO3 loss on inducing the phenotypic changes of lung fibroblasts, we performed specific knockdown of FoxO3 in N-HLF using siRNA (Fig 2A; Appendix Fig S9). Interestingly, knockdown of FoxO3 significantly increased the proliferation,

both in basal and medium containing FCS or in the presence of IGF-I or PDGF-BB, when compared to scramble siRNA-transfected cells (Fig 2B). Notably, the expression of fibroblast-to-myofibroblast differentiation markers (*COL1A1*, *COL3A1,* and *ACTA2*; Fig 2C) and transcriptional co-factor, *Myocardin* (Appendix Fig S10), was significantly increased after FoxO3 knockdown compared to control. These findings led us to analyze its function on the fibrotic process *in vivo*.

### Global- and fibroblast-specific FoxO3 knockout mice aggravate bleomycin-induced pulmonary fibrosis

To define the role of FoxO3 on the fibrotic processes *in vivo*, we employed the classical and well-characterized bleomycin-induced pulmonary fibrosis mouse model. We first determined whether FoxO3 is also dysregulated during the fibrogenic phase of bleomycin-induced lung injury. Similar to IPF fibroblasts, FoxO3 mRNA and protein levels were significantly decreased in lung fibroblasts isolated from bleomycin-treated mice 14 or 21 days post-instillation

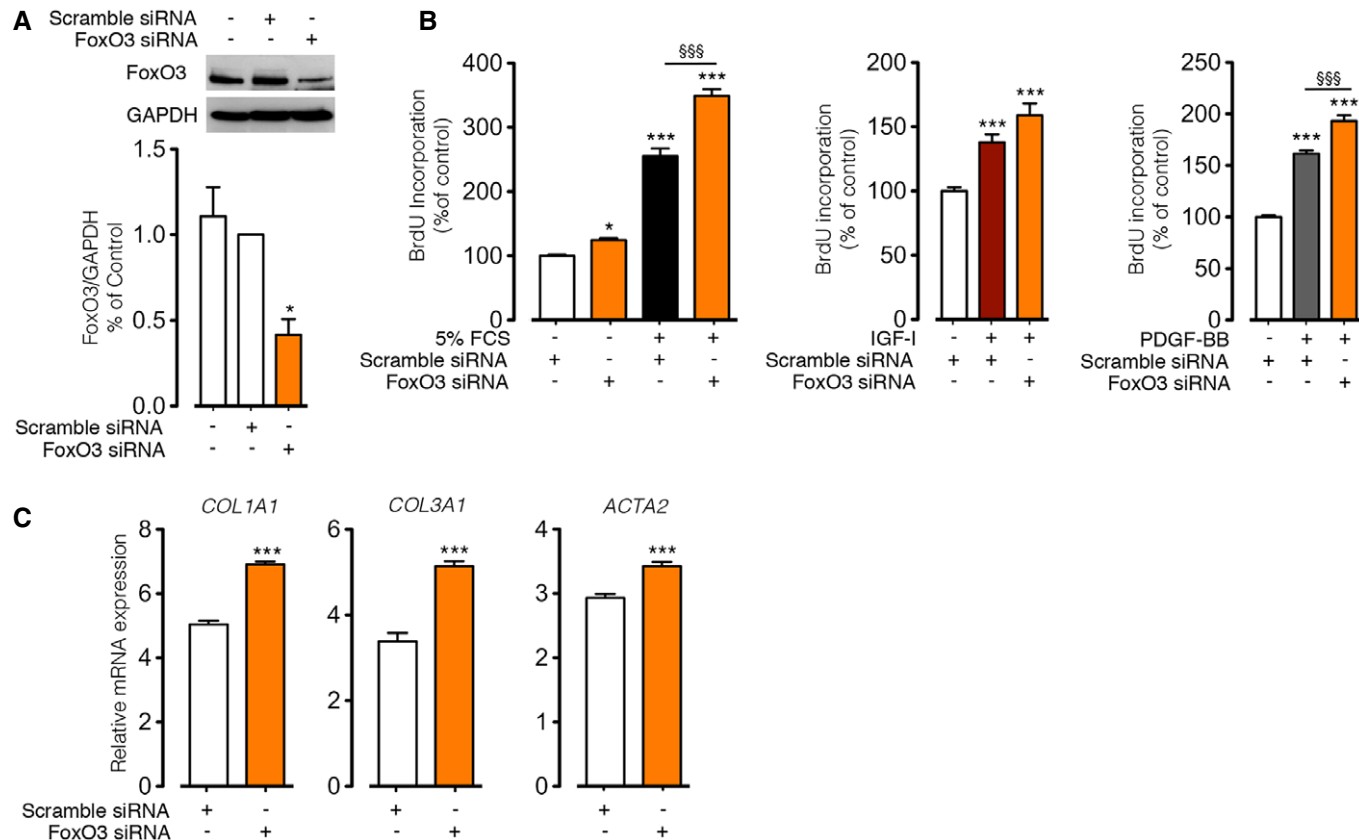

**Figure 2.  FoxO3 knockdown in control human lung fibroblasts mimics the pro-proliferative and myofibroblast phenotypes.**

A        Representative Western blots of FoxO3 in control human lung fibroblasts untransfected or transfected with scramble siRNA or FoxO3 siRNA. Densitometry quantified data of FoxO3/GAPDH expression ratio (*n* = 3).

B, C   Human lung fibroblasts from donors (*n* = 2–3) were transfected with scramble siRNA or FoxO3 siRNA. (B) Transfected cells were stimulated with 5% FCS, IGF-I (200 ng/ml), and PDGF-BB (60 ng/ml) or left non-stimulated, and cell proliferation was measured by BrdU incorporation. (C) mRNA expression of fibroblasts to myofibroblasts markers (*ACTA2*, *COL1A1*, and *COL3A1*) was analyzed by qPCR.

Data information: Data are expressed as mean ± SEM. In (A and B), data are represented as a percentage of control, non-stimulated scramble siRNA-transfected cells. In (A), data were analyzed using repeated-measures one-way ANOVA. In (B), data were analyzed using one-way ANOVA. In (C), data were analyzed using Student's *t*-test. **P* < 0.05, ****P* < 0.001 versus non-stimulated scramble siRNA; $^{§§§}P$ < 0.001 versus stimulated scramble siRNA.

                                                                    

when compared to saline-instilled mice (Fig 3A and B). Furthermore, the level of inactive FoxO3, p-FoxO3, was significantly increased in lung fibroblasts of bleomycin-treated mice at 14 and 21 days post-instillation in comparison with the controls (Fig 3B). These data strongly support the association between FoxO3 downregulation and processes of myofibroblast activation and fibrosis after lung injury.

Next, we examined the role of FoxO3 *in vivo* using global FoxO3 knockout mice ($Foxo3^{-/-}$) and mice lacking FoxO3 in fibroblasts ($Foxo3_{f.b}^{-/-}$). Fibroblast-specific FoxO3 knockout mice ($Foxo3_{f.b}^{-/-}$) were generated by crossing a floxed allele of *Foxo3* with mice expressing Cre under the control of *Col1a1* promoter. Fibroblast specificity of $Foxo3_{f.b}^{-/-}$ was confirmed at mRNA level (Appendix Fig S11A and B) and in lung sections by

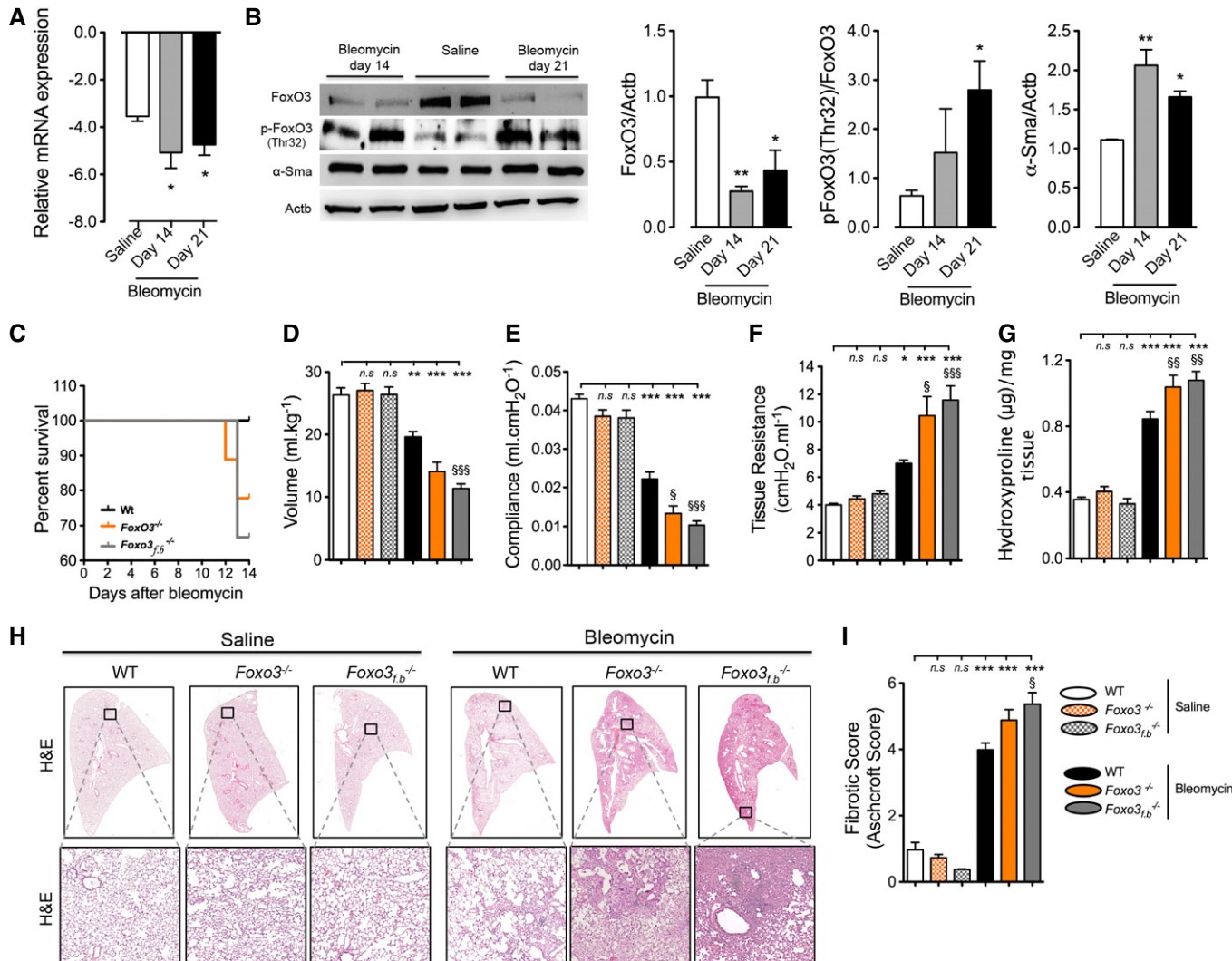

**Figure 3.  Aggravation of bleomycin-induced lung fibrosis in mice with global- and fibroblast-specific Foxo3-knockout mice.**

A, B    Fibroblasts were isolated from saline-treated mice lungs at day 21 post-instillation and mice lungs after 14 or 21 days of bleomycin instillation. (A) mRNA expression analysis of *Foxo3* by qPCR (n = 4–5/group). (B, left panel) Representative Western blots of p-FoxO3 (Thr32), FoxO3, and α-Sma protein levels. Actb was used as a loading control. (B, right panel) Densitometry quantified data of FoxO3, p-FoxO3 Thr32, and α-Sma-to-Actb expression ratio (n = 4/group).

C    Percentage of survival of mice challenged with bleomycin. WT (littermates) bleomycin (n = 6 mice), $Foxo3_{f.b}^{-/-}$ (n = 6 mice) and $Foxo3^{-/-}$ bleomycin (n = 9 mice).

D–F    Lung function measurements of mice at day 14 after instillation, (D) total lung capacity, (E) lung compliance, and (F) lung tissue resistance.

G    Hydroxyproline levels in mouse lungs.

H    Representative H&E staining of whole left lung (upper panel) and higher magnification (lower panel). Scale bar = 100 μm.

I    Fibrotic score.

Data information: Data are expressed as mean ± SEM. In (D–I) WT saline (n = 7 mice), $Foxo3^{-/-}$ (n = 6 mice), $Foxo3_{f.b}^{-/-}$ (n = 5 mice), WT bleomycin (n = 6 mice), $Foxo3_{f.b}^{-/-}$ bleomycin (n = 4 mice), and $Foxo3^{-/-}$ bleomycin (n = 7 mice). Data were analyzed using one-way ANOVA. *P < 0.05, **P < 0.01, ***P < 0.001 versus WT saline. §P < 0.05, §§P < 0.01, §§§P < 0.001 versus WT bleomycin.

immunofluorescence (Appendix Fig S11C). All mutant Foxo3 mice ($Foxo3^{-/-}$, $Foxo3_{f.b}^{-/-}$) did not exhibit any overt phenotype and had a body weight similar to that of wild-type littermates. $Foxo3^{-/-}$ and $Foxo3_{f.b}^{-/-}$ animals and WT littermates were orotracheally instilled with bleomycin (day 0). As a control, $Foxo3^{-/-}$ and WT mice were instilled with saline. None of saline-treated mice died (data not shown). On the other hand, $Foxo3^{-/-}$ and $Foxo3_{f.b}^{-/-}$ bleomycin-instilled mice showed a decrease in survival at day 12 of post-bleomycin instillation, whereas no death in WT mice was observed at this time point (Fig 3C).

Due to the high mortality and decrease in body weight observed in Foxo3 mutant bleomycin-treated mice, the experiment was terminated at day 14 and lung function measurements were performed. Foxo3 deletion did not affect lung functions properties as $Foxo3^{-/-}$ saline-instilled mice had a total lung capacity, lung compliance, and tissue resistance similar to that of WT saline-treated mice. In contrast, $Foxo3^{-/-}$ and $Foxo3_{f.b}^{-/-}$ mice challenged with bleomycin had significantly impaired lung function compared to WT bleomycin-instilled mice. As shown in Fig 3D and E, total lung capacity and lung compliance of $Foxo3^{-/-}$ and $Foxo3_{f.b}^{-/-}$-bleomycin-injured mice were significantly decreased in comparison with WT bleomycin-instilled mice. Furthermore, their lung tissue resistance was significantly increased as compared to WT bleomycin-instilled mice (Fig 3F). In agreement with these observations, FoxO3 depletion accompanied the fibrotic response to bleomycin, as determined by histopathology and assessment of fibrotic score and hydroxyproline. $Foxo3^{-/-}$ and $Foxo3_{f.b}^{-/-}$ bleomycin-instilled mouse lung sections demonstrated a significant increase in fibrotic score and hydroxyproline levels as compared to WT littermates (Fig 3G–I). To study the inflammatory responses, we performed immunohistochemical staining for CD68 (macrophages), CD3 (T cells), and CD45 (leukocytes). We observed a marked infiltration of CD68-, CD3-, and CD45-positive cells, localized to sites of injury and fibrosis within the lungs of bleomycin-challenged WT, $Foxo3^{-/-}$ and $Foxo3_{f.b}^{-/-}$ lungs of mice (Fig EV3A–C). Interestingly, an augmented increase in CD3-positive cells was found in both $Foxo3^{-/-}$ and $Foxo3_{f.b}^{-/-}$ mouse lungs compared to WT mouse lungs (Fig EV3D). Infiltration of T cells in global- and fibroblast-specific FoxO3 knockout mice is intriguing and may suggest that FoxO3 depletion in fibroblasts may promote recruitment of T cells to the fibrotic area by secreting chemokines and promote chronic inflammation.

In summary, these results indicate that FoxO3 has an important role in attenuating the severity of lung fibrosis induced by bleomycin instillation.

## UCN-01 inhibits pro-fibrotic factors driven fibroblast differentiation and proliferation

The results presented above prompted us to hypothesize that reactivation of FoxO3 may provide a novel therapeutic option for IPF disease. To test this hypothesis, a virtual screening and extensive search was performed in a database of compounds, which have been shown to increase FoxO3 transactivation, at low concentration, either by inhibiting FoxO3 phosphorylation or nuclear export or by increasing FoxO3 expression. This screen identified UCN-01, 7-hydroxystaurosporine, as a potential candidate in terms of its abilities to inhibit FoxO3 phosphorylation and translocation and supported by the fact that this agent is approved for phase II

anti-cancer clinical trials (http://clinicaltrials.gov). Thus, we assessed potential effects of UCN-01 on lung fibroblast proliferation and transdifferentiation induced by fibrotic factors. N-HLF was stimulated with 5% FCS and treated with increasing concentrations of UCN-01 determined by MTT assay (Appendix Fig S12). UCN-01 (50, 100, or 200 nM) significantly decreased proliferation by ≈30% in comparison with vehicle treatment (Fig 4A). Similar to FCS, UCN-01 significantly inhibited N-HLF proliferation induced by PDGF-BB and IGF-I (Fig 4B and C).

To assess the effect of UCN-01 on fibroblast transdifferentiation induced by TGF-β1, N-HLF were stimulated with TGF-β1 for 24 h in the absence or presence of UCN-01. TGF-β1 treatment significantly increased mRNA levels of myofibroblast differentiation markers (COL1A1, COL3A1, and ACTA2) as compared to untreated cells (Fig 4D). Protein levels of Col1a and α-SMA were also significantly increased after TGF-β1 treatment (Fig 4E). Importantly, pretreatment with UCN-01 strongly inhibited TGF-β1-induced mRNA expression of COL1A1, COL3A1, and ACTA2. Consistent with the RNA analysis, the protein levels of Col1a and α-SMA were also decreased by UCN-01 treatment (Fig 4E and F; protein quantification and immunostainings).

## UCN-01 inhibits FoxO phosphorylation and translocation in growth factor-stimulated N-HLF via PI3K/Akt

To determine whether the anti-proliferative effect of UCN-01 is accompanied by increased FoxO3 activity, we assessed the status of FoxO3 phosphorylation and nucleus–cytoplasm translocation in UCN-01-treated N-HLF in the presence of pro-fibrotic factors (FCS, PDGF-BB, IGF-1). Treatment of N-HLF with UCN-01 for 30 min significantly reduced FoxO3 phosphorylation at $Thr^{32}$ and $Ser^{253}$ in response to FCS, PDGF-BB, or IGF-I stimulation (Fig 5A–I). In addition, UCN-01 treatment inhibited FoxO3 nuclear exclusion in N-HLF induced by FCS, PDGF-BB, and IGF-I (Fig 5J, Appendix Fig S13).

We next explored the molecular basis of decreased phosphorylation and nuclear retention of FoxO3 in FCS-, PDGF-BB-, and IGF-I-stimulated N-HLF in the presence of UCN-01. Since the FoxO3 phosphorylation at $Thr^{32}$ and $Ser^{253}$ observed in this study can be mediated by activated PI3K/Akt pathway (Brunet et al, 1999), we examined the PI3K/Akt pathway involvement in the UCN-01 inhibited FoxO3 phosphorylation. Consistent with these notions, an increase in p-AKT was observed in FCS-, PDGF-BB-, and IGF-I-stimulated N-HLF, which was diminished by UCN-01 (Fig 5A–I). A similar effect was observed when cells were pretreated with wortmannin, a well-known PI3K inhibitor (Fig 5A–J). Collectively, these data suggest that UCN-01 blocks FoxO3 phosphorylation and thereby causes its nuclear retention in primary human lung fibroblasts by inhibiting Akt activation by FCS, PDGF-BB, and IGF-I.

## UCN-01 increases FoxO expression and inhibits PI3K/Akt-dependent FoxO phosphorylation in TGF-β1-stimulated N-HLF

Next, we assessed whether UCN-01 inhibition of TGF-β1-induced myofibroblast differentiation is also associated with FoxO3 regulation. Treatment of donor lung fibroblasts with UCN-01 for 24 h significantly inhibited TGF-β1-induced FoxO3 downregulation at mRNA and protein levels (Fig 6A and B). In addition, UCN-01 treatment significantly reduced the levels of phosphorylated FoxO3 at

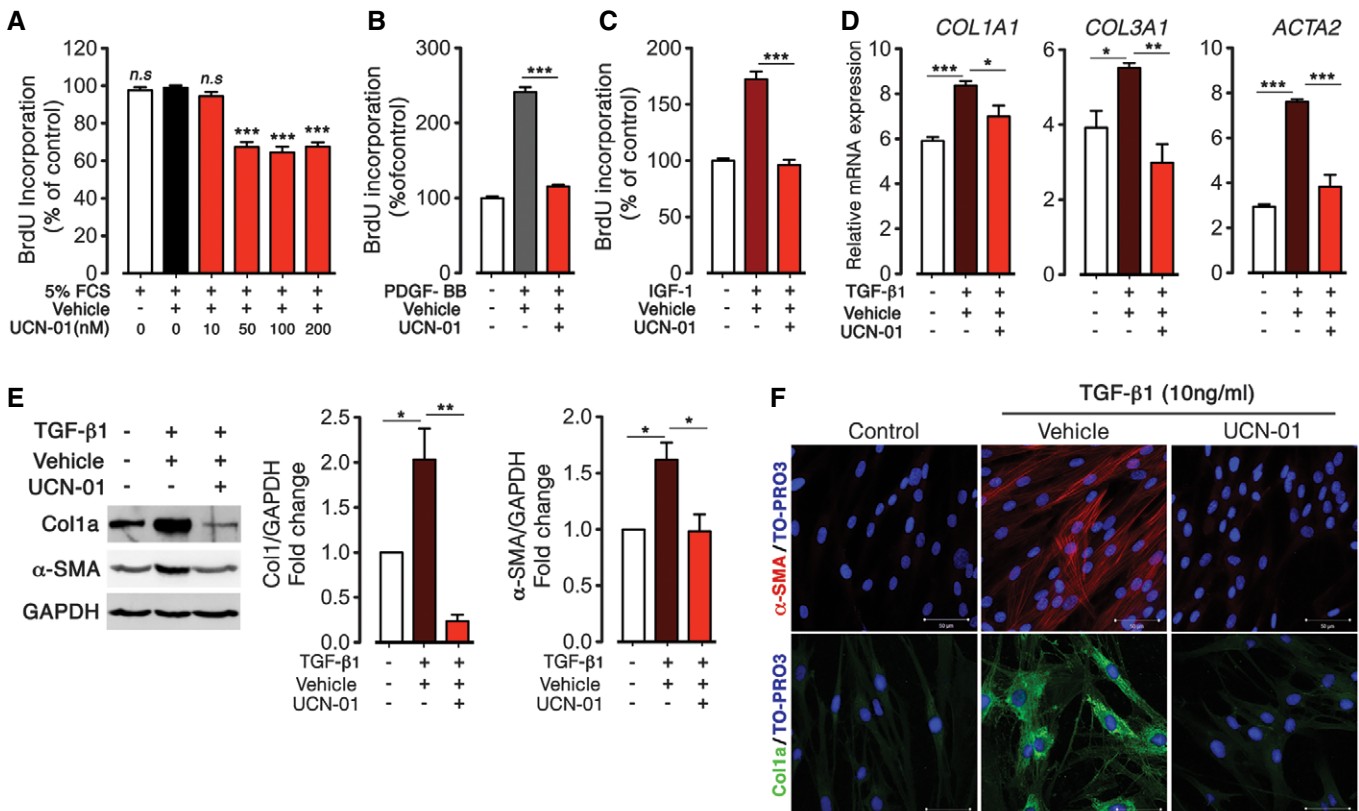

**Figure 4.  UCN-O1 inhibits human lung fibroblasts proliferation and fibroblast-to-myofibroblasts transdifferentiation in response to pro-fibrotic stimuli.**

A–C   Serum-starved (48 h) N-HLF (n = 3) were stimulated with 5% FCS (A), PDGF-BB (60 ng/ml) (B), or IGF-1 (200 ng/ml) (C) in the presence of UCN-01 or vehicle (DMSO) as indicated and cell proliferation was measured by BrdU incorporation after 24 h. Data represent percentage of control; vehicle-treated cells (A) or serum-starved cells (B and C).

D–F   Serum-starved (48 h) N-HLF (n = 3) were left non-stimulated or stimulated with TGF-β1 (10 ng/ml) in medium containing UCN-01 (50 nM) or vehicle (DMSO) for an additional 24 h and the following analyses were performed: (D) mRNA expression analysis of fibroblast-to-myofibroblasts markers (COL1A1, COL3A1, and ACTA2) expression by qPCR. (E, left panel) Representative Western blots of collagen 1a (Col1a) and α-SMA. GAPDH was used as a loading control. (E, right panel) Densitometry quantified data of Col1a and α-SMA-to-GAPDH expression ratio. (F) ICC of α-SMA (red, upper) and Col1a (green, lower). TO-PRO3 (blue) was used to label nuclei. Scale bar = 50 μm.

Data information: Data are expressed as mean ± SEM. In (A–D), data were analyzed using one-way ANOVA, n.s. = not significant, *P < 0.05, **P < 0.01, ***P < 0.001 versus vehicle-treated cells. Data in (E) were analyzed using repeated-measures one-way ANOVA, *P < 0.05, ***P < 0.01 versus vehicle-treated cells.

Thr$^{32}$ and Ser$^{253}$. This effect was associated with a decrease in phosphorylated AKT, albeit with no changes in total AKT (Fig 6B and D). Similar effect was observed when cells were pretreated with wortmannin (Appendix Fig S14). Together, these data indicate that UCN-01 increases FoxO3 level in TGF-β1-stimulated primary human lung fibroblasts by inhibiting AKT activation.

**Anti-proliferative and anti-differentiation effects of UCN-01 on primary lung fibroblasts are largely mediated via FoxO3**

To assess whether the UCN-01 effects is mediated via FoxO3, we downregulated FoxO3 and treated N-HLF with UCN-01 in the presence of mitotic stimulus. Downregulation of FoxO3 significantly reduced the anti-proliferative effect of UCN-01 (Fig 7A). Interestingly, the percentage, ~30%, of the increase in proliferation by FoxO3 siRNA was similar to the percentage of reduction in the anti-proliferative effect of UCN-01 after using FoxO3 siRNA (Fig 7A). Similarly, knockdown of FoxO3 decreased the inhibitory effect of

UCN-01 on TGF-β1-induced myofibroblast differentiation markers (COL1A1, COL3A1, and ACTA2; Fig 7B).

**UCN-01 inhibits IPF-HLF myofibroblast differentiation and proliferation**

To ascertain the therapeutic potential of FoxO3 activity modulation by UCN-01 in bona fide diseased lung fibroblasts, we performed corresponding studies in primary lung fibroblasts isolated from IPF patients, IPF-HLF. Similar to the effects observed in N-HLF, UCN-01 inhibited FoxO3 nuclear exclusion of IPF fibroblasts induced by FCS, PDGF-BB, or IGF-1 (Fig EV4). In accordance, UCN-01 significantly inhibited IPF-HLF proliferation in a dose-dependent manner (Fig 8A). Furthermore, UCN-01 inhibited AKT phosphorylation, FoxO3 phosphorylation, and FoxO3 reduction, in TGF-β1-stimulated IPF-HLF (Fig 8B). This effect was accompanied by a marked decrease in the expression of myofibroblast differentiation markers, Col1a and α-SMA (Fig 8B).

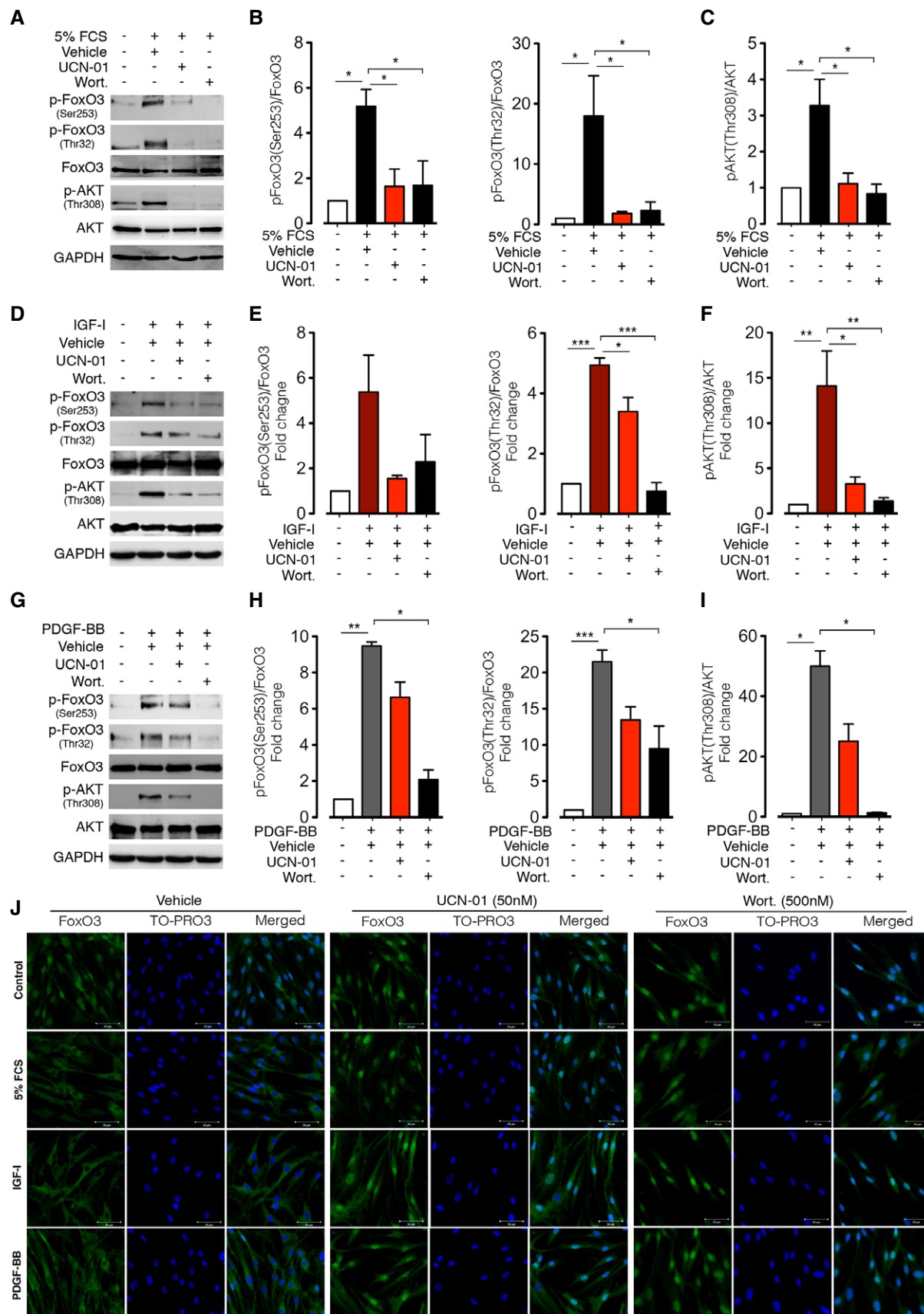

**Figure 5.**

**Figure 5.  UCN-01 inhibits FoxO3 inactivation in human lung fibroblasts stimulated with growth factors.**

A–I   Serum-starved (48 h) N-HLF were stimulated with 5% FCS (A–C), IGF-1 (200 ng/ml) (D–F), or PDGF-BB (60 ng/ml) (G–I) in medium containing UCN-01 (50 nM) or wortmannin (Wort.) (500 nM) or vehicle (DMSO) for 30 min and Western blot analysis was performed with the antibodies as described. (A, D, and G) Representative Western blots of p-FoxO3 (Thr32), p-FoxO3 (Ser253), FoxO3, p-AKT (Thr308), AKT, and GAPDH. (B, C, E, F, H, and I) Densitometry quantified data of p-FoxO3 (Thr32) or p-FoxO3 (Ser253) to FoxO3 and p-AKT (Thr308) to AKT expression ratios, represented as a fold change to non-stimulated cells.

J     ICC of FoxO3 in N-HLF that were serum-starved for 48 h and were stimulated with 5% FCS or PDGF-BB (60 ng/ml) or IGF-1 (200 ng/ml) and treated with 50 nM UCN-01 or 500 nM wortmannin or vehicle control (DMSO) for 6 h. TO-PRO3 (blue) was used to label nuclei. FoxO3 and TO-PRO3 images were overlaid to visualize nuclear and cytoplasmic localization of FoxO3. Images are representative of $n = 3$. Scale bar = 50 μm.

Data information: Data are expressed as mean ± SEM and were analyzed using repeated-measures one-way ANOVA, *$P < 0.05$, **$P < 0.01$, ***$P < 0.001$ versus vehicle-treated cells.

Source data are available online for this figure.

## UCN-01 abrogates fibrogenic responses in the bleomycin-induced mouse model pulmonary fibrosis

To determine the therapeutic potential of UCN-01 *in vivo*, WT mice were treated with UCN-01 (5 or 7.5 mg/kg) or vehicle on every second day starting at day 7 after the bleomycin challenge and analyzed for the therapeutic benefit of UCN-01 at day 21 (Fig 8C). The choice of day 7 as beginning of the treatment is in line with protocols studying therapeutic approaches in this model (Moeller *et al*, 2008). We observed no difference in lung

function measurements between vehicle-treated and untreated bleomycin mice groups (Fig 8D–F). In contrast, the lung function of UCN-01-treated mice was improved in a dose-dependent manner in comparison with the vehicle-treated mice, as demonstrated by a significant increase in total lung capacity and lung compliance, and by a significant decrease in tissue resistance with no change in inflammatory cell composition (Fig 8D–F, Appendix Fig S15A–D). Consistent with the lung function measurement, analysis of lung sections revealed a significant decrease in fibrotic score and collagen deposition in

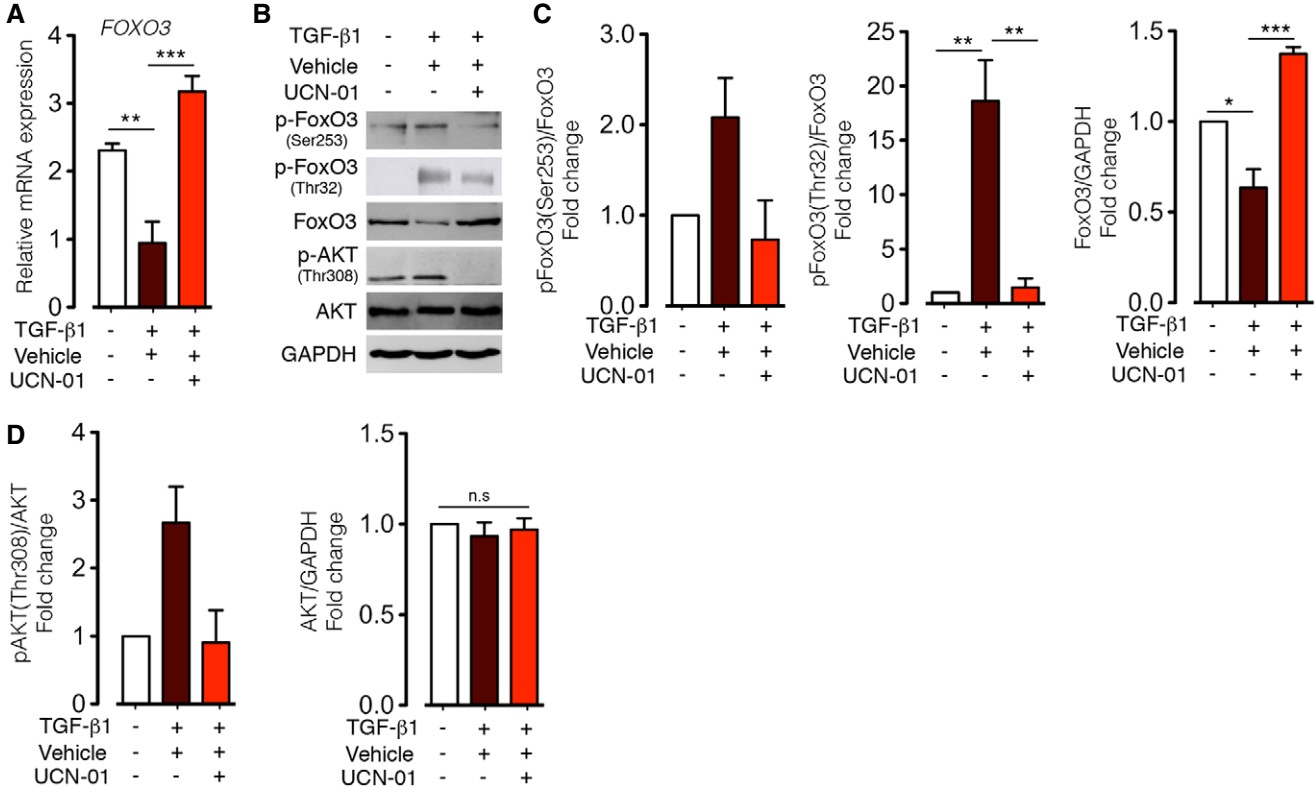

**Figure 6.  UCN-01 inhibits FoxO3 inactivation in human lung fibroblasts stimulated with TGF-β1.**

A–D   Forty-eight hours serum-starved N-HLF were left non-stimulated or stimulated with TGF-β1 (10 ng/ml) in medium containing UCN-01 (50 nM) or vehicle (DMSO) for 24 h. (A) mRNA expression of *FOXO3* by qPCR. Data are expressed as mean ± SEM and were analyzed using one-way ANOVA, **$P < 0.01$, ***$P < 0.001$ versus vehicle-treated cells. (B) Representative Western blots of p-FoxO3 (Thr32), p-FoxO3 (Ser253), FoxO3, p-AKT (Thr308), AKT and GAPDH. (C, D) Densitometry quantified data, represented as fold change ($n = 3$). In all panels, cells were serum-starved for 48 h before stimulations. Data are expressed as mean ± SEM and were analyzed using repeated-measures one-way ANOVA, n.s. = not significant, *$P < 0.05$, **$P < 0.01$, ***$P < 0.001$ versus vehicle-treated cells.

Source data are available online for this figure.

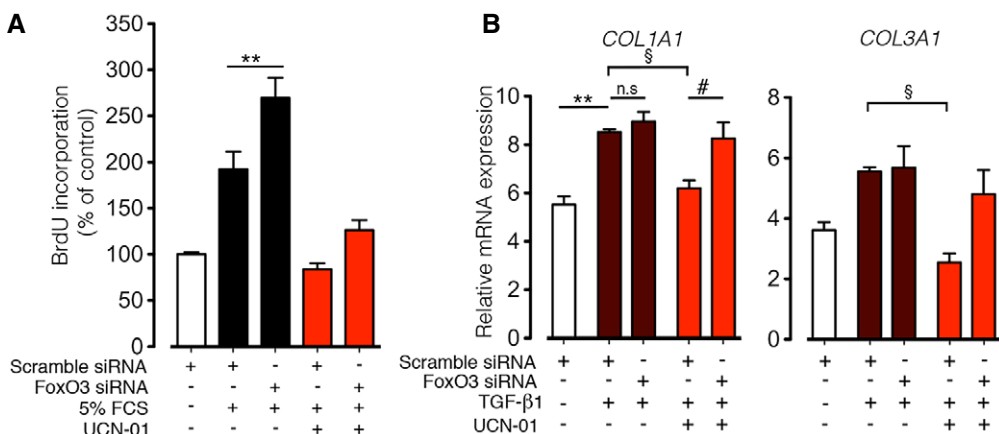

**Figure 7. Preceding knockdown of FoxO3 reduces inhibitory effect of UCN-01 on human lung fibroblast proliferation and myofibroblast transdifferentiation in response to pro-fibrotic stimuli.**

A N-HLF were transfected with scramble siRNA or FoxO3 siRNA. 6 h after transfection, cells were serum-starved for 36 h and then stimulated with 5% FCS in the presence or absence of UCN-01 (50 nM), and cell proliferation was measured by BrdU incorporation after 24 h. Scramble siRNA-transfected cells were left non-stimulated for 24 h as an additional control. Data represent percentage of control, scramble siRNA non-stimulated cells ($n$ = 2–3). Data are expressed as mean $\pm$ SEM and were analyzed using one-way ANOVA, **$P$ < 0.01 versus 5% FCS-scramble siRNA.

B N-HLF was transfected with scramble siRNA or FoxO3 siRNA. 6 h after transfection, cells were serum-starved for 36 h and then stimulated with TGF-β1 (10 ng/ml) in the presence or absence of UCN-01 (50 nM) for 24 h and mRNA expression of *COL1A1* and *COL3A1* were measured with qPCR ($n$ = 3). Data are expressed as mean $\pm$ SEM and were analyzed using one-way ANOVA, n.s. = not significant, **$P$ < 0.01 versus scramble siRNA, §$P$ < 0.05 versus TGF-β1-scramble siRNA, #$P$ < 0.05 versus UCN-01-scramble siRNA.

UCN-01-treated mice as compared to vehicle-treated mice (Fig 8G and H, and Appendix Fig S15E).

### Anti-fibrotic effect of UCN-01 is mediated via FoxO3 *in vivo* and *in vitro*

To investigate whether the effect of UCN-01 on lung function and fibrosis is regulated via FoxO3, *Foxo3*$^{-/-}$ mice and WT littermates were treated with UCN-01 (7.5 mg/kg) or vehicle on every second day starting at day 7 after bleomycin challenge and analyzed for therapeutic benefit of UCN-01 at day 21. In WT bleomycin-challenged mice, UCN-01 treatment improved total lung capacity and lung compliance, and a significant decrease in tissue resistance and hydroxyproline content as compared to vehicle-treated WT mice (Fig 9). On the other hand, *Foxo3*$^{-/-}$ mice challenged with bleomycin had significantly impaired lung function compared to WT bleomycin-instilled mice. Importantly, even in these *Foxo3*$^{-/-}$ mice with aggravated bleomycin-induced pulmonary fibrosis, UCN-01 failed to rescue fibrosis. FoxO3 depletion decreased the beneficial effects of UCN-01 more than 50% on total lung capacity and lung compliance (Fig 9A and B). Similarly, it also partly failed to rescue lung tissue resistance and fibrosis (Fig 9C and D).

To analyze how much anti-proliferative and anti-fibrotic effect of UCN-01 is mediated by selectively inhibiting FoxO3 phosphorylation downstream of PDK1/PI3K/AKT, we overexpressed constitutively active AKT deletion mutant (PH domain deletion mimics constant phosphorylation, hence, constant activation) in combination with UCN-01 under 5% FCS and TGF-β stimulation. We observed that constitutive activation of AKT reversed the decrease observed in FoxO3 phosphorylation induced by UCN-01 under both 5% FCS (Fig EV5A) and TGF-β (Fig 9E) to a substantial extent. This clearly indicates that UCN-01 decreases FoxO3 phosphorylation by

inhibiting PDK1/AKT pathway. Additionally, constitutive active AKT was able to reverse the anti-proliferative effect of UCN-01 under 5% FCS stimulation substantially (Fig EV5B). Interestingly, it was also able to significantly increase the expression of fibrotic markers (Col1A1, Col3A1) that was decreased by UCN-01 treatment with TGF-β stimulation (Fig 9F). These findings undoubtedly signify that majority of effect of UCN-01 proliferation, and fibrosis is mediated via PDK1/AKT-controlled FoxO3 phosphorylation.

## Discussion

The present study, performed in human *ex vivo*-cultured fibroblasts of healthy and IPF lung origin and a mouse model of lung fibrosis with employment of knockout animals, forwarded strong evidence for a crucial role of FoxO3 in integrating various growth factor signaling pathways and controlling downstream gene expression (Fig 9G). Loss of nuclear FoxO3 activity due to downregulation of its message and/or phosphorylation with subsequent nuclear exclusion caused phenotypic changes of the fibroblasts with enhanced proliferation and differentiation to myofibroblasts. In corroboration, enhanced lung fibrosis, loss of lung function, and reduced survival were observed in the mouse fibrosis model. Importantly, employing a novel pharmacological tool for nuclear FoxO3 reconstitution blocked the fibroblast phenotypic change *in vitro* and rescued lung function *in vivo* (Fig 9G). These studies implicate a critical inhibitory role of FoxO3 in IPF fibrogenesis and proof of concept for therapeutic reactivation of FoxO3 as an effective therapeutic strategy for pulmonary fibrosis.

The defining pathological feature of IPF is the formation of fibroblastic foci, which are areas of accumulated fibroblasts/myofibroblasts within collagen-rich matrix. The formation and

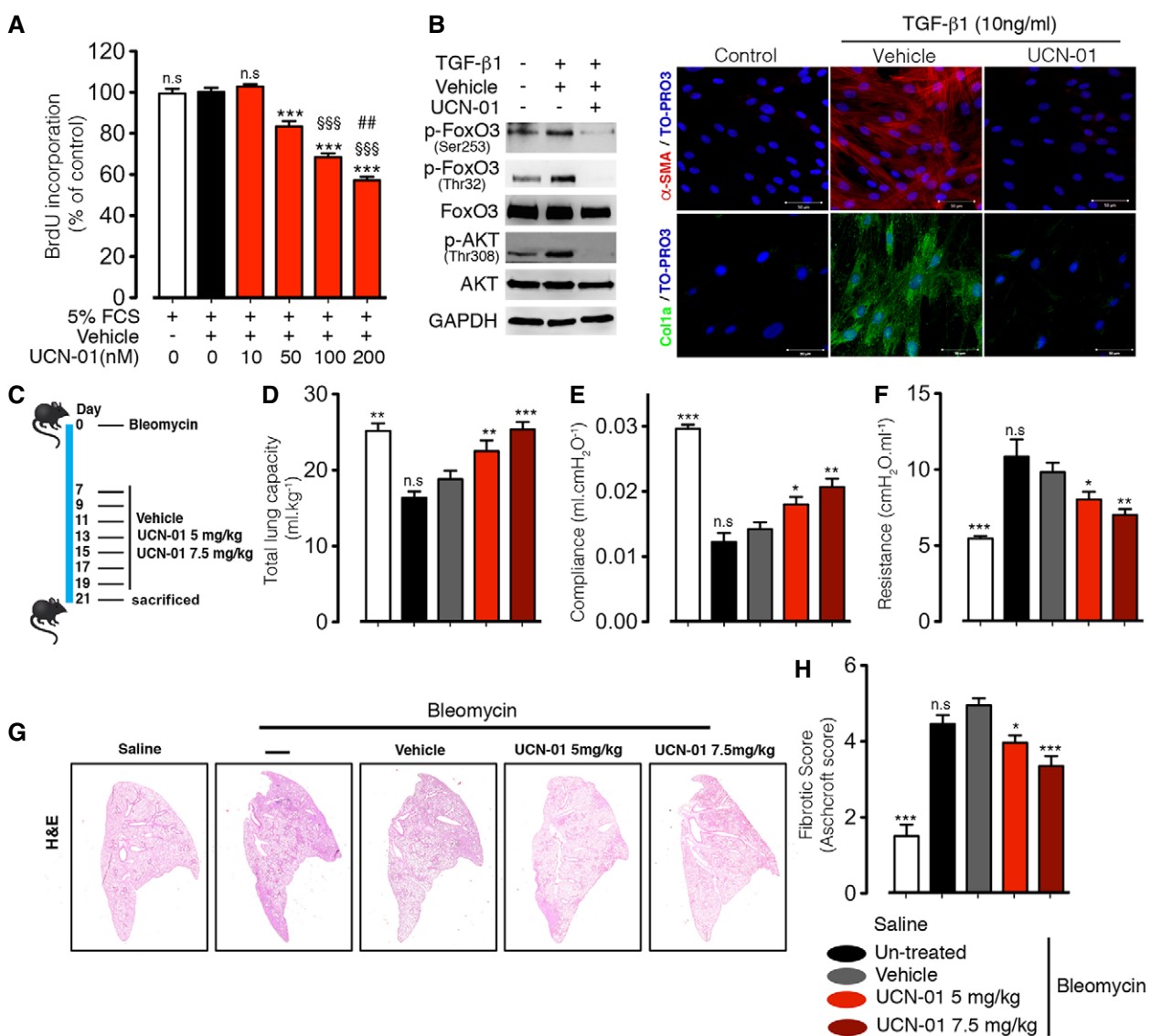

**Figure 8.   UCN-01 inhibits human IPF lung fibroblast pathological phenotypes and attenuates lung fibrosis induced by bleomycin injury.**

A    Serum-starved (48 h) IPF-HLF ($n$ = 5) were stimulated with 5% FCS and treated with UCN-01 (10, 50 100, and 200 nM) or vehicle (DMSO) or left untreated, and cell proliferation was measured by BrdU incorporation after 24 h.

B    IPF-HLF cells that were serum-starved for 48 h and stimulated with TGF-β1 (10 ng/ml) and treated with UCN-01 (50 nM) or vehicle (DMSO) for 24 h. (B, left panel) Western blots of p-FoxO3 (Thr32), p-FoxO3 (Ser253), FoxO3, p-AKT (Thr308), AKT, and GAPDH. (B, right panel) ICC of α-SMA (red) and Col1a (green). TO-PRO3 (blue) was used to label nuclei. Scale bar = 50 μm. Images are representative of $n$ = 3.

C    Scheme shows experimental setup.

D–F  Lung function measurements of mice, (D) total lung capacity, (E) lung compliance, and (F) tissue resistance.

G    Representative H&E staining of whole left lung.

H    Fibrotic score.

Data information: Data are expressed as mean ± SEM. Data in (A) were analyzed using one-way ANOVA, n.s. = not significant, ***$P$ < 0.001 versus vehicle-treated cells, §§§$P$ < 0.001 versus 50 nM UCN-01-treated cells, ##$P$ < 0.01 versus 100 nM UCN-01-treated cells. In (C–H): saline ($n$ = 8 mice), bleomycin non-treated ($n$ = 7 mice), bleomycin vehicle-treated ($n$ = 8 mice), UCN-01 5 mg/kg.bw ($n$ = 10 mice), and UCN-01 7.5 mg/kg.bw ($n$ = 7 mice). In (D–F and H), data were analyzed using one-way ANOVA, n.s. = not significant, *$P$ < 0.05, **$P$ < 0.01,***$P$ < 0.001 versus vehicle-treated mice.

Source data are available online for this figure.

progressive expansion of fibroblastic foci in the lung parenchyma results in the destruction of alveolar architecture and impairment of gas exchange (King *et al*, 2011). Not surprisingly, the number of fibroblast foci per field has been shown to be prognostically relevant (Harada *et al*, 2013). Thus, activated (myo)fibroblasts, displaying a different phenotype as compared to normal lung fibroblasts, are key players, driving the relentless progression of IPF (Xia *et al*, 2008). Consistent with this notion, an increased rate of proliferation and augmented myofibroblast differentiation markers were observed in the fibroblasts isolated from IPF lungs—as compared to control

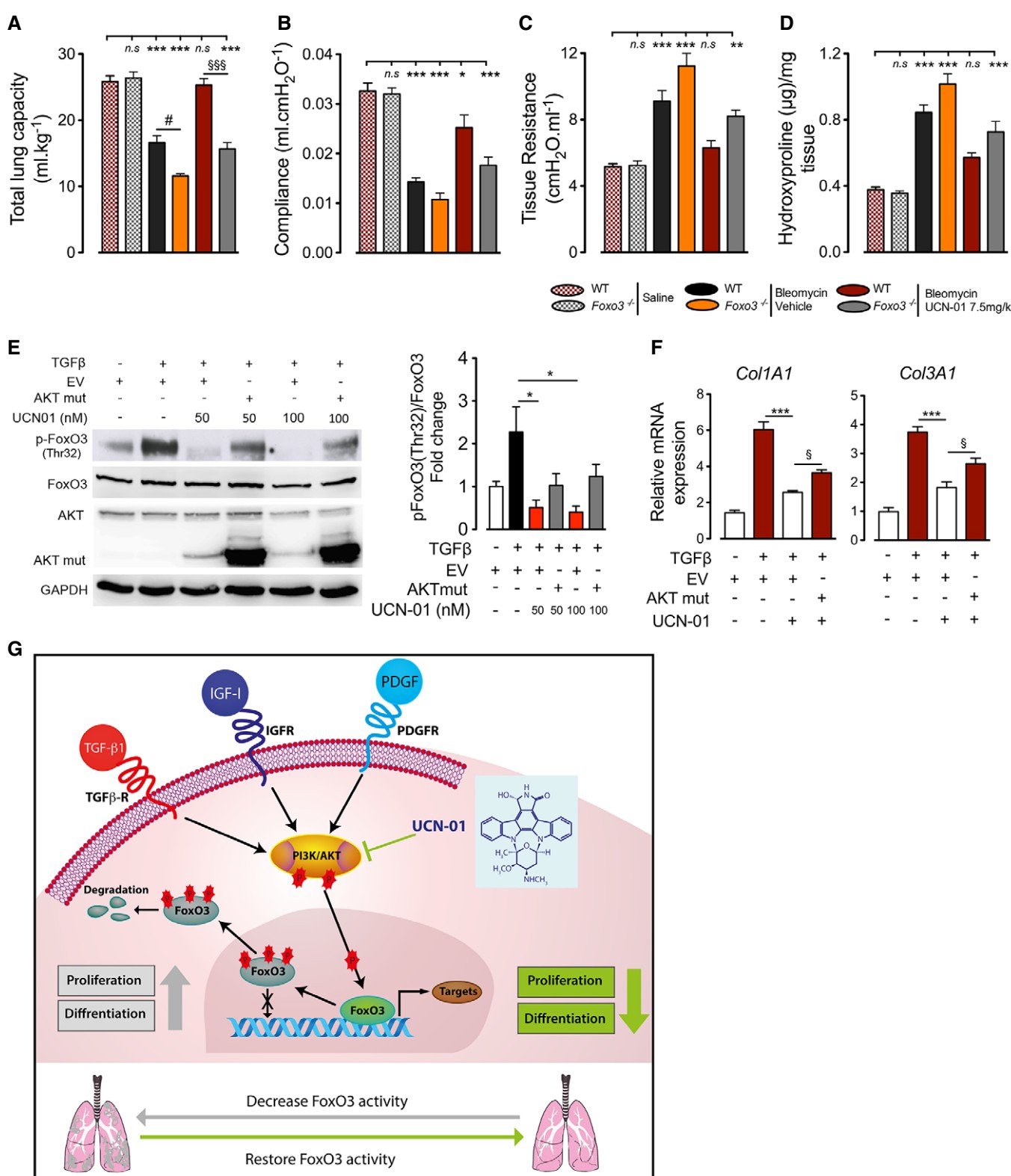

Figure 9.

healthy donor lungs—in the current study. Such "activated myofibroblast state" is maintained *ex vivo*, at least up to 10 passages in which these cells were used. Importantly, a reduced expression, as well as inactivation due to increased phosphorylation of FoxO3, was consistently observed in the human IPF fibroblasts as well as in fibroblasts derived from the bleomycin model of pulmonary fibrosis.

◄

**Figure 9.  UCN-01 inhibits lung fibrosis via regulation of FoxO3.**

A–C    Lung function measurements of mice, (A) total lung capacity, (B) lung compliance, and (C) tissue resistance.
D       Hydroxyproline levels in mice lungs.
E, F    Stimulation with TGF-β1 (10 ng/ml) in the presence or absence of UCN-01 (50 nM) for 24 h, followed by (E) Western blotting for p-FoxO3 (Thr32), FoxO3, AKT, GAPDH, and (F) qPCRs for *COL1A1* and *COL3A1* (n = 3).
G       Model of signaling events involving FoxO3 during IPF pathogenesis. In lung fibroblasts, FoxO3 with nuclear localization leads to transcription of genes involved in regulation of apoptosis, migration, differentiation, and cell-cycle inhibition. Pro-fibrotic factors (TGF-β1, IGF-1, PDGF-BB) bind to their receptors and activate PI3K, followed by phosphorylation of AKT. AKT phosphorylates and thereby inactivates FoxO3, followed by nuclear exclusion and degradation of FoxO3, resulting in progression of lung fibrosis. UCN-01 inhibits AKT phosphorylation resulting in FoxO3 reactivation and attenuation of disease progression. Transforming growth factor β receptor (TGF-βR), transforming growth factor β1 (TGF-β1), platelet-derived growth factor receptor (PDGFR), platelet-derived growth factor-BB (PDGF-BB), insulin-like growth factor receptor (IGFR), and insulin-like growth factor-1 (IGF-1).

Data information: Data are expressed as mean ± SEM. In (A–C), WT saline (n = 13 mice), *Foxo3*$^{-/-}$ saline mice (n = 6 mice), WT bleomycin vehicle-treated (n = 14 mice), *Foxo3*$^{-/-}$ bleomycin vehicle-treated mice (n = 7 mice), WT bleomycin-UCN-01 7.5 mg/kg.bw (n = 11 mice), *Foxo3*$^{-/-}$ bleomycin-UCN-01 7.5 mg/kg.bw mice (n = 5 mice). (D) Collagen deposition: WT saline (n = 4 mice), *Foxo3*$^{-/-}$ saline mice (n = 6 mice), WT bleomycin vehicle-treated (n = 5 mice), *Foxo3*$^{-/-}$ bleomycin vehicle-treated mice (n = 6 mice), WT bleomycin-UCN-01 7.5 mg/kg.bw (n = 5 mice), *Foxo3*$^{-/-}$ bleomycin-UCN-01 7.5 mg/kg.bw mice (n = 5 mice). Data in (A–D) were analyzed using one-way ANOVA, *$P < 0.05$, **$P < 0.01$, and ***$P < 0.001$ versus WT saline mice; #$P < 0.05$ versus WT bleomycin-UCN-01 mice; §§§$P < 0.001$ versus WT bleomycin-UCN-01 mice. Data in (E and F) were analyzed using one-way ANOVA, *$P < 0.05$, ***$P < 0.001$ versus TGF-β1-EV and §$P < 0.05$ versus UCN-01-EV.

This observation is consistent with a previous report, describing a decrease in total as well as nuclear FoxO3 protein levels and an increase in phosphorylated FoxO3 in IPF fibrotic areas (Nho *et al*, 2011). Against the background that FoxO transcription factors serve as central regulators of gene expression, this observation suggests a causal relationship between FoxO3 loss and the fibroblast phenotypic change.

In addition, our findings supported the notion that FoxO3 acts as a critical downstream integrator of several growth factor pathways (FCS, PDGF-BB, IGF-1) in human lung fibroblasts, the growth factor pathways that were known to mimic pro-fibrotic environment (6,13,17), linked with the phenotypical change of these cells. PDGF-BB is known to signal via binding to its receptors PDGF-Rα and/or PDGF-Rβ, causing receptor dimerization and phosphorylation, which results in the activation of different intracellular signaling pathways (Bonner, 2004). Surprisingly, to the best of our knowledge, the activated intracellular signaling pathways by which PDGF ligands induce lung fibroblast proliferation have not been reported. However, several *in vitro* studies, carried out in myofibroblast-like cells involved in other fibrotic diseases including hepatic stellate cells (HSC) and renal mesangial cells, revealed that PDGF-BB induces proliferation via activation of the PI3K-AKT pathway (Reif *et al*, 2003; Adachi *et al*, 2007). FCS was similarly reported to induce HSC proliferation via PI3K-AKT (Reif *et al*, 2003; Adachi *et al*, 2007). IGF-1 binds to its receptor IGF-1R, resulting in IGF-1R autophosphorylation, which triggers a cascade of intracellular signaling. IGF-1 was also shown to activate PI3K/AKT in primary mouse lung fibroblasts (Choi *et al*, 2009). Consistent with these reports, stimulation of human lung fibroblasts with FCS, PDGF-BB, and IGF-1 was found to result in PI3K/AKT pathway activation in the present study, as evidenced by the increased level of phosphorylated AKT, which was inhibited by pretreatment with a PI3K inhibitor. Active AKT is known to phosphorylate FoxO3 at Thr$^{32}$ and Ser$^{253}$, as demonstrated here for both *ex vivo*-isolated human IPF fibroblasts and for control human lung fibroblasts exposed to pro-proliferative growth factors *in vitro* (Brunet *et al*, 1999; Dobson *et al*, 2011).

TGF-β1 is known as major fibrogenic factor and was shown to promote lung fibroblast-to-myofibroblasts transdifferentiation, characterized by increased a-SMA expression and ECM protein synthesis

including Col1a1 and Col3a1 (Raghu *et al*, 1989; Wang *et al*, 2006; Gauldie *et al*, 2007; Kulkarni *et al*, 2011). However, the molecular pathways involved in TGF-β1-induced fibroblast transdifferentiation have only partially been identified. Next to the classical TGF-β1 signaling engaging SMAD2/3, SMAD-independent pathways including PI3K/AKT have been shown to take a part in TGF-β1-induced mesenchymal transition (Wang *et al*, 2006; Kolosionek *et al*, 2009; Conte *et al*, 2011, 2013; Kulkarni *et al*, 2011). The downstream target/s of the TGF-β1-activated PI3K/AKT are, however, not reported for fibroblasts. This study provides evidence that FoxO3 is one of these missing downstream molecules. Stimulation of human lung fibroblasts with TGF-β1 for 24 h induced AKT activation as evidenced by increased phosphorylated AKT level. This induction was associated with increased FoxO3 phosphorylation at Thr$^{32}$ and Ser$^{253}$ (AKT-dependent phosphorylation sites) and downregulation of FoxO3, suggesting that FoxO3 activity is interlinked with fibroblast-to-myofibroblast transdifferentiation. In line with this suggestion, we found that FoxO3 activity is regulated during fibroblast-to-myofibroblast transformation.

Of note, TGF-β1 stimulation demonstrated a different pattern of FoxO3 nuclear exclusion compared to PDGF-BB, FCS, and IGF-1. TGF-β1 stimulation did not result in a clear FoxO3 nuclear exclusion up to 24 h, though TGF-β1 leads to FoxO3 phosphorylation and AKT activation within 2 h. We assume that interference or counter-regulation by other activated pathways upon TGF-β1 stimulation may result in a slower kinetics of nuclear exclusion of phosphorylated FoxO3 (Meyer *et al*, 2012), which requires further investigation.

In line with the notion that FoxO3 serves as a critical downstream integrator of several growth factor pathways in human lung fibroblasts and that its downregulation/inactivation/nuclear exclusion is linked with the phenotypical change of these cells, FoxO3 knockdown in control human lung fibroblasts mimicked the major features of this phenotypical change, causing increased proliferation as well as enhanced expression of fibroblast-to-myofibroblast transdifferentiation markers (*COL1A1*, *COL3A1*, and *ACTA2*). Interestingly, increased expression of these markers was also associated with higher expression of MYOCARDIN, a co-transcription factor shown to upregulate expression of myofibroblast markers in hepatic stellate cells during liver fibrosis (Shimada & Rajagopalan, 2012),

hence, indicating role of myocardin in mediating effect of FoxO3 in regulating myofibroblast differentiation in pulmonary fibrosis. Moreover, most of these *in vitro* findings were recapitulated *in vivo* in bleomycin-challenged global-, as well as fibroblast-specific FoxO3 knockout mice. These mouse lines displayed markedly aggravated pulmonary fibrosis as compared to wild-type controls. In particular in the specific $Foxo3_{f,b}^{-/-}$ knockout line, a relentless, progressive feature of the fibrosis was noted, along with severely impaired lung function parameters (lung volume, lung compliance, tissue resistance), augmented fibrosis score, and early death of the animals. To the best of our knowledge, this is the first study to definitively implicate fibroblast-specific FoxO3 in fibrogenesis. Collectively, these *in vitro* and *in vivo* data indicate that loss of FoxO3 in lung fibroblasts is critically involved in the progression of pulmonary fibrosis.

These findings prompted the reasoning that reactivation of FoxO3 might be an effective novel strategy for blocking lung fibrogenesis. However, despite broad interest in FoxO reactivation in the cancer field—impaired FoxO activity has been reported in many cancer cell lines (Hu *et al*, 2004; Savai *et al*, 2014)—so far the only efficient method to increase FoxO activity is by using inhibitors that block upstream regulators of FoxO. As we explored that enhanced PI3K/AKT pathway activity was the main driver of FoxO3 inactivation in lung fibroblasts, a search for pharmacological inhibition of this pathway was performed, forwarding UCN-01 (7-hydroxystaurosporine) as agent of interest. UCN-01 is currently being tested in several clinical trials in the cancer field, as single agent or in combination with other agents (http://clinicaltrials.gov). This derivative of the serine/threonine kinase inhibitor staurosporine inhibits PDK1, a kinase crucial for PI3K/AKT signaling (Sato *et al*, 2002; Komander *et al*, 2003): activated P13K/AKT generates phosphatidylinositol 3,4,5-triphosphate, which in turn recruits AKT to the plasma membrane where AKT is phosphorylated (activated) on two key regulatory sites, at Ser[473] by mTOR (Sarbassov *et al*, 2005) and at Thr[308] by PDK1 (Stephens *et al*, 1998). Phosphorylation at both sites is necessary for full activation of AKT and its downstream effects (Hemmings & Restuccia, 2012). Our studies add to this body of knowledge that UCN-01 blocks AKT phosphorylation at Thr[308] with subsequent reduction in FoxO3 phosphorylation. UCN-01 exhibits anti-cancer effects on different cancer cells, including those of lung, colon, and breast origin, both *in vivo* and *in vitro* (Shao *et al*, 1997; Usuda *et al*, 2000).

Employing UCN-01 to activate FoxO3 in lung fibrosis, this agent was found to inhibit proliferation of IPF fibroblasts as well as donor fibroblasts that were stimulated by FCS, PDGF-BB, or IGF-1. Furthermore, UCN-01 inhibited fibroblast-to-myofibroblast transdifferentiation induced by TGF-β1. This effect was accompanied by reconstitution of FoxO3 activity, as a decrease in FoxO3 phosphorylation and an increase in FoxO3 expression were observed. Importantly, downregulation of FoxO3 by FoxO3 siRNA significantly reduced the anti-proliferative and anti-differentiation potential of UCN-01, proving that the effect of this agent is to a major extent mediated via FoxO3-dependent gene regulation. Employing UCN-01 as a therapeutic agent in the bleomycin model, applied 7 days after onset of injury, where transition from inflammation to fibrosis occurs, significantly attenuated lung fibrosis, as evidenced by markedly improved lung function parameters and lung structure, and improved survival. Most importantly, UCN-01 was unable to rescue

fibrosis to a considerable extent in $Foxo3^{-/-}$ bleomycin-treated mice, strongly supporting the mediation of anti-fibrotic effect of UCN-01 via FoxO3.

In conclusion, we observed that FoxO3 is a critical integrator of various growth factor signaling pathways and plays a crucial role in suppression of the phenotypic change from fibroblasts to activated myofibroblasts. Reactivation of FoxO3 by UCN-01 reversed the phenotypic change and reverted pulmonary fibrosis. Re-purposing of UCN-01 for the treatment of IPF may offer as novel therapeutic option for this devastating disease.

# Materials and Methods

## Human primary lung fibroblasts

All biomaterials were provided by the UGMLC Giessen Biobank, member of the DZL Platform Biobanking. Informed consent was obtained from all the subjects and experiments conformed to the principles set out in the WMA Declaration of Helsinki and the Department of Health and Human Services Belmont Report. Control primary human lung fibroblasts (N-HLF) were derived from healthy donor lung tissue. Control lung tissue demonstrated anatomically normal lung structure. IPF primary human lung fibroblasts (IPF-HLFs) were derived from IPF patients' lung tissue. The diagnosis of IPF was confirmed by histological analysis of lung tissue and exhibited the characteristic morphological findings of usual interstitial pneumonia. The institutional ethics committee approved the protocol and tissue usage, and written informed consent was obtained from all patients before the procedures being performed (Pullamsetti *et al*, 2011; Savai *et al*, 2014). Lung fibroblasts were generated by explant method. Nine N-HLF and seven IPF-HLF fibroblasts were characterized as previously described and used in this study (Chen *et al*, 1992). The isolated fibroblasts were maintained at 37°C and 5%$O_2$ in humidified chamber, and cultured in 100-mm plates in growth medium (MCDB-131 medium containing 5% FCS, 1% L-glutamine, penicillin (100 U/ml), streptomycin (0.1 mg/ml), EGF (0.5 ng/ml), bFGF 2 ng/ml, and insulin (5 μg/ml). At 90–95% confluence, cells were subcultured. Because of the concern that the phenotype of the cells is altered at higher passage, cells between passages 4 and 6 were utilized in the experiments described here, except cells for screening were used at earlier passages.

## Mice primary cell isolation

### Fibroblasts

Primary mouse lung fibroblasts were isolated from 10-week-old wild-type FVB/N mice. Mice lungs were perfused with PBS. Lungs were then washed with ice-cold HBSS-Ca/Mg and lungs tissues were minced with sterilized scissors. Minced tissues were then incubated on shaker at 37°C for 1 h with digestion buffer (0.3 mg/ml type IV collagenase and 0.5 mg/ml trypsin dissolved in HBSS-Ca/Mg). Digested tissues were then passed through 40-μm cell strainer. Cells were centrifuged at 800 rpm for 5 min, and cell pellets were resuspended in 9 ml growth medium (DMEM/F12 containing 10% FCS, 1% L-glutamine, penicillin (100 U/ml), and streptomycin (0.1 mg/ml) and seeded in a culture plates. Since fibroblast cells are known to attach very fast in comparison with epithelial cells, seeded cells

were left to attach for 2–4 h. Next, cultured plates were washed with PBS and a fresh growth medium was added. At 80% confluency, cells were splitted. The purity of isolated fibroblasts was verified by positive staining for vimentin, fibronectin, and collagen I, as well as negative staining for pro-surfactant protein C. Mouse lung fibroblasts were used between passages 3 and 6 (Pullamsetti *et al*, 2011).

### Alveolar type II cells

Lung cell suspensions were prepared by intratracheal instillation of dispase and agarose followed by mechanical disaggregation of the lungs. Crude cell suspensions were purified by negative selection using a biotinylated antibody, streptavidin-coated biomagnetic particle system as described in previous report (Zhou *et al*, 2015). The purity of isolated type II cells was verified by positive staining for pro-surfactant protein C and negative staining for vimentin and collagen I.

## Stimulation and treatment of lung fibroblasts

Stimulation and treatment of human lung fibroblasts were carried out in serum starvation conditions. Serum starvation of human lung fibroblast was performed using MCDB-131 medium containing 1% L-glutamine, penicillin (100 U/ml), and streptomycin (0.1 mg/ml). Serum starvation of mouse lung fibroblasts was performed using DMEM/F12 containing 1% L-glutamine, penicillin (100 U/ml), and streptomycin (0.1 mg/ml). Primary mouse lung fibroblasts were isolated from 10-week-old FVB/N mice. Recombinant human IGF-1 and recombinant human PDGF-BB were purchased from Panbiotech, USA, and recombinant human TGF-β1 from R&D, USA. UCN-01 (Sigma, USA) and wortmannin (Calbiochem, Germany) were dissolved in dimethyl sulfoxide (DMSO).

## Quantitative real-time PCR (qPCR)

RNA isolation was carried out using TRIzol™ reagent. After DNase1, cDNA was synthesized using ImProm-II™ Reverse transcription system (Promega, USA). mRNA expression analysis was performed using Brilliant III ultra-fast SYBR®Green qPCR master mix (Agilent Technologies, USA). Expression levels are relative to HPRT (HPRT-gene). Primers sequences are listed in Appendix Table S1.

## Western blotting and quantification

Lung tissue samples and cells were homogenized in lysis buffer, and protein lysates were separated on 10% SDS–Poly Acrylamide gels and transferred to nitrocellulose membrane. After blocking, the membranes were incubated with primary antibodies (Appendix Table S2) over night and followed by 1-h incubation with horseradish peroxidase-conjugated secondary antibodies. Chemiluminescent signal was detected by exposing the membrane to films. Films were developed using Curix 60 (AGFA, Belgium) and scanned using BioDoc from Biometra, and densitometric quantification of bands was performed using BioDoc Analyze software from (Biometra). In some experiments, chemiluminescent signal was detected using Image reader of Fujifilm LAS 4000 and densitometric quantification of bands was measured using multigauge software (Fujifilm).

## Fibroblast proliferation assays

Lung fibroblasts, to be assessed for cellular proliferation, were cultured either in 96-well or 48-well plates. Fibroblast proliferation was determined using colorimetric BrdU incorporation assay kit (Roche, Germany) according to manufacturer's instructions. Absorbance was measured at 370 nm with reference at 492 nm in a plate reader (TECAN, Germany). Depending on the experiment, proliferation of cells was plotted either as the difference of absorbance at 370 and 492 nm (A370 nm–A492 nm) or as a percentage of absorbance compared to control cells absorbance.

## Immunocytochemistry (ICC)

Human lung fibroblasts to be subjected to ICC were cultured on eight-well glass chamber slides. Cells were fixed with ice-cold methanol/acetone (1:1) for 30 min, washed three times for 5 min with PBS and blocked for 1 h with blocking buffer (5% BSA, 0.5% goat serum, 0.2% Triton-X in PBS), and incubated over night with one of the following antibodies: Col1a (1:100, Median life science, T40777R), FoxO3 (1:100, cell signalling, 2497), and α-SMACy3 (1:200 Sigma Aldrich, C6198). This was followed by 1-h incubation with secondary antibody Alexa Fluor®-488 (1:1,000, Life Technologies, A11008) except for the cells that were probed with α-SMA. TOPO3 was used for nuclear counter stain. Fluorescent images were taken with LSM 710 confocal microscope.

## Fibroblast transfection

Human lung fibroblasts were transfected using Amaxa™ Basic Nucleofector™ primary fibroblast kit (Lonza, USA). Briefly, $1 \times 10^6$ cells were resuspended in nucleofection buffer. Then, FoxO3 siRNA or scramble siRNA (invitogen) was added and mixed with the cells. Cells were then electroporated using Amaxa™ Nucleofector II™ system (Lonza, USA), resuspended in medium and cultured.

## Fibrosis score and Collagen content

Fibrosis score was assessed using the Ashcroft score as previously described (Pullamsetti *et al*, 2011); numerically scale the grade (from 0 to 8) of fibrotic changes in histological lung samples. Sections (4 μm) of the left lobe of mice lungs were stained with H&E. Whole sections were scanned with 10-fold magnification using Leica DM6000B microscope (Leica, Germany). This resulted in around 25–40 pictures per mouse lung. The pictures from each mouse lung were graded, averaged, and depict in a graph. The whole-lung lobe pictures were scanned with Nanozoomer 2.0 HT (Hamamatsu, Japan). Collagen content was assessed by sirius red staining of lung sections, and the area of collagen was measured using Leica image software (Leica Microsystems, Germany) according to the standard protocol.

## Hydroxyproline measurements

Hydroxyproline levels in murine lung tissue were determined following the protocol of Woessner (Woessner, 1961) using the QuickZyme Hydroxyproline Assay kit (Quickzyme Biosciences, Leiden, the Netherlands). One lobe from the right lung was taken

for analysis and the weight determined. The lung tissue was homogenized in 1 ml 6 N HCl with a Precellys tissue homogenizer (2 × 20 s, 3,800 *g*). The homogenate was then hydrolyzed at 90°C for 24 h. After centrifugation at 13,000 *g* for 10 min, 100 µl from the supernatant was taken and diluted 1:2 with 4 N HCl. 35 µl of this working dilution was transferred to a 96-well plate. Likewise, a hydroxyproline standard (12.5–300 µM) was prepared in 4 N HCl and transferred to the microtiterplate. Following addition of 75 µl of a chloramine T-containing assay buffer, samples were oxidized for 20 min at room temperature. The detection reagent containing p-dimethylaminobenzaldehyde was prepared according to the manufacturer's instruction and 75 µl added to the wells. After incubation at 60°C for 1 h, the absorbance was read at 570 nm with a microtiter plate reader (Infinite M200 Pro, Tecan, Crailsheim, Germany) and the hydroxyproline concentration in the sample was calculated from the standard curve and related to the employed amount of lung tissue. The hydroxyproline content in lung tissue is given as µg hydroxyproline per mg lung tissue.

## Animal experiments

The experiments were performed in accordance with the US National Institutes of Health Guidelines on the Use of Laboratory Animals. Both the University Animal Care Committee and the federal authorities for animal research of the Regierungspräsidium Darmstadt (Hessen, Germany) approved the study protocol (approval numbers B2/299; B2/309; B2/359). Mice with the null (-) and floxed allele (L) for *Foxo3* were kindly provided by Prof. Ronald A. Depinho (Harvard Medical School, Boston, USA). To specifically delete Foxo3 in fibroblasts ($Foxo3_{f.b}^{-/-}$), homozygous floxed mice for *Foxo3* allele ($Foxo3^{L/L}$) were crossed with transgenic mice expressing Cre under the control of collagen type 1A1 promoter (Col1a1-Cre). $Foxo3_{f.b}^{+/-}$ mice were intercrossed, and animals homozygous for Foxo3 deletion in fibroblasts ($Foxo3_{f.b}^{-/-}$) were obtained. Mice littermates were used as WT control. Col1a1-Cre transgenic mice were purchased from Mutant Mouse Regional Resource Centers (MMRRC; stock ID: 000208-UCD). 8- to 10-week-old male mice were orotracheally instilled with 2.5 U/kg body weight (in UCN-01 experiment) or 3.5 U/kg body weight (in transgenic experiment) of bleomycin (Medac, Germany) that was dissolved in saline (Pullamsetti *et al*, 2011). Control groups were instilled with saline.

## Lung function measurements

Lung function measurements were obtained using FlexiVentTM system (Scireq, Canada) equipped with FlexiVent 7 operating software. Mice were mechanically ventilated at a rate of 150 breaths/min, tidal volume of 10 ml/kg, and a positive end-expiratory pressure (PEEP) of 3 cmH$_2$O for 3 min. Deep inflation perturbation (slow inflation of the lung to 30 cmH$_2$O) was initiated to measure the total lung capacity. Snapshot perturbation maneuver was imposed to measure the lung compliance, which is obtained by fitting the maneuver signals to a single compartment model of the lung by the operating software. This perturbation was repeated for 3–4 times every 30 s. Then, broadband low-frequency forced oscillation measurements (1–20.5 Hz) were performed using "Quick Prime-3 perturbation". The resulting respiratory input impedance

## The paper explained

### Problem

Idiopathic pulmonary fibrosis (IPF) is a progressive and fatal parenchymal lung disease with limited therapeutic options and unclear signaling mechanisms. It has been recognized that in IPF, fibroblasts are phenotypically altered to a highly contractile and synthetic fibroblast phenotype (myofibroblasts). The molecular changes involved in these disordered functions of IPF fibroblasts have not been elucidated.

### Results

In this study, we show that FoxO3 is a crucial player in driving IPF fibroblast phenotype by performing experiments in human IPF tissue, human *ex vivo*-cultured fibroblasts of healthy and IPF lung origin and a mouse model of lung fibrosis with employment of global- and fibroblast-specific FoxO3 knockout mice. Importantly, we provide evidence for UCN-01, a staurosporine derivative, as a novel therapeutic approach to treat IPF.

### Impact

Therapy targeting activation of FoxO3 hence holds high translational potency for a currently devastating disease.

was fit to the constant phase model of the lung by the operating software, and lung tissue resistance was obtained. Similarly to snapshot perturbation, Quick Prime-3 perturbation was repeated three to four times every 30 s. A coefficient of determination of 0.95 was the lower limit for accepting perturbation outcomes.

## Cell count in bronchoalveolar lavage fluid

Lungs were lavaged with three sequential 300 µl of PBS. Recovered BALF was centrifuged, and the pellet was resuspended in saline solution. Cells were spun onto a slide in a Cytospin 3 centrifuge, stained with May Grunwald/Giemsa staining, and counted under light microscope as previously described (Pullamsetti *et al*, 2011).

## Statistical analysis

All data are expressed as mean ± standard error of the mean (SEM). Statistical comparisons of samples were performed by Student's *t*-test for comparing two groups or repeated-measure ANOVA or one-way ANOVA followed by the Tukey's *post hoc* test for multiple comparisons. Difference with $P < 0.05$ between the groups was considered significant. Exact *P*-values for the statistical analysis are provided in Appendix Table S3.

**Expanded View** for this article is available online.

## Acknowledgements

We acknowledge excellent technical assistance of Uta Eule in bleomycin animal experiments, Katharina Leib for cell culture experiments, Marianne Hoeck and Natascha Wilker for BAL analysis and histological studies. This work was supported by the Max Planck Society, Excellence Cluster Cardio-Pulmonarfigy System (ECCPS), Verein zur Förderung der Krebsforschung in Gießen e.V., a Von-Behring-Röntgen-Stiftung Grant, a Rhön Klinikum AG Grant, a LOEWE Universities of Giessen and Marburg Lung Center Grant, German Center for Lung Research (DZL) and SFB 1213 project A01.

## Author contributions

HMA-T, RS, and SSP conceived and designed research study. HMA-T, AS, SD, CR, and PS acquired the data. HMA-T, JP, and RAD contributed transgenic mice. HMA-T, AS, SD, RS, and SSP analyzed and interpreted the data. HMA-T, AG, OE, WS, and SSP drafted the manuscript. SD and SSP revised the manuscript. RS, FG, WS, and SSP handled the funding and supervision.

## Conflict of interest

The authors declare that they have no conflict of interest.

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
