## [Review Process File · EMBO Molecular Medicine]

FoxO3 an important player in fibrogenesis and therapeutic target for idiopathic pulmonary fibrosis

Hamza M. Al-Tamari, Swati Dabral, Anja Schmall, Pouya Sarvari, Clemens Ruppert, Jihye Paik, Ronald A. DePinho, Friedrich Grimminger, Oliver Eickelberg, Andreas Guenther, Werner Seeger, Rajkumar Savai and Soni S. Pullamsetti

Corresponding author: Soni S. Pullamsetti, Max-Planck-Institute for Heart and Lung Research

Review timeline:	Submission date:	29 January 2016
	Editorial Decision:	14 March 2016
	Additional correspondence (author):	27 July 2016
	Revision received:	08 August 2017
	Editorial Decision:	06 October 2017
	Revision received:	25 November 2017
	Accepted:	03 November 2017

Transaction Report:

Editors: Roberto Buccione, Céline Carret

1st Editorial Decision

14 March 2016

Thank you for the submission of your manuscript to EMBO Molecular Medicine.

I am sorry for the belated decision but in this case we experienced unusual difficulties in securing three willing and appropriate reviewers. Furthermore and unfortunately, quite late into the evaluation process one reviewer (#2) made him/herself unavailable. As a further delay cannot be justified I have now decided to proceed based on the two available consistent evaluations.

Both Reviewers are mostly positive on your manuscript although they raise some very important issues that require your action. I will not dwell into the specifics, as their comments are very detailed and self-explanatory. I would like, however, to highlight a few main points.

Reviewer 1 as you will see, has some concerns with respect to overall novelty, which is exacerbated by the perceived lack of conclusiveness and experimental support for the main conclusions. This includes among other things, that s/he is unconvinced that the differences in FoxO3 expression and phosphorylation are strictly IPF-related and would like to see confirmation from patient tissue analysis. Reviewer 1 also argues that a better assessment of PF is needed in the mouse model and suggests that lung hydroxyproline levels would be the better endpoint in this case. Finally, s/he notes that UCN-01 (and wortmannin) is far from being a specific kinase inhibitor and thus off-targets effects cannot be excluded. The reviewer also lists several other items of concern for your action.

Reviewer 3 raises somewhat similar concerns including on novelty. On one hand s/he is not convinced that the impact of FoxO3 in PF is fibroblast specific. On the other, reviewer 3, as reviewer 1, is also concerned that UCN-01 is too unspecific to allow you to draw specific conclusions on the kinase.

In conclusion, while publication of the paper cannot be considered at this stage, we would be willing to consider a substantially revised submission, with the understanding that the all Reviewers'

concerns must be addressed with additional experimentation where appropriate and that acceptance of the manuscript will entail a second round of review. This is especially important given that the required experimentation would consolidate the clinical relevance and originality of the work being presented and would significantly upgrade the relevance and conclusiveness of the dataset.

I understand that if you do not have the required data available at least in part, to address the above, this might entail a significant amount of time, additional work and experimentation and might be technically challenging, I would therefore understand if you chose to rather seek publication elsewhere at this stage. Should you do so, we would welcome a message to this effect.

I look forward to seeing a revised form of your manuscript as soon as possible.

***** Reviewer's comments *****

Referee #1 (Comments on Novelty/Model System):

The manuscript will require more attention being paid to making sure key pharmacological agents used to probe different pathways, including in particular data generated with UCN-1 and wortmannin are on-target. Detailed concentration-response curves are needed as detailed in the remarks below. The endpoint for assessing fibrosis should also include assessment of lung hydroxyproline levels.

Referee #1 (Remarks):

General comments:

This study is based within the clinical setting of idiopathic pulmonary fibrosis, and evaluates the potential therapeutic benefit of preventing the phosphorylation and consequent inactivation of FoxO3 by PI3K/Akt, which has previously been implicated in mediating the fibroproliferative response in this condition. The observation that this PI3K/Akt - FoxO3 axis may play an important role in this disease setting is not novel and has been extensively investigated. Indeed some of the key initial datasets are largely confirmatory of existing literature. This article does however extend current knowledge in particular with respect to the potential importance of this pathway in the context of experimentally-induced lung fibrosis. A particularly attractive aspect of the study relates to a potential therapeutic approach which could potentially be taken forward into clinical testing. However, there are several critical issues which require attention, including the need to pay greater attention to concentration-responses with broad-spectrum pharmacological agents and the choice of final endpoints to assess the fibrotic response in the animal model.

Major comments:

1. Fig 1D and 1E are critical in that they set the premise for all the ensuing work addressing the importance of FoxO3 as a key regulator of fibroblast responses in IPF. Regrettably, the data presented are not particularly convincing in terms of reporting key differences between the levels of FoxO3 expression in IPF versus non-IPF fibroblast lines (Fig 1D) in that the phosphorylation state of FoxO3 appears to be highly heterogeneous in the IPF lines with only 2/4 showing high levels of phosphorylation. This reviewer is therefore not convinced that these data support the conclusion that this is unique to IPF and would argue that fibroblast heterogeneity might provide explain the reported differences across donor and IPF lines. Confirmatory studies of FoxO3 expression and phosphorylation status based on detailed analysis of patient biopsy tissue are absolutely essential to support the current proposition that this is an important disease-relevant and targetable pathway.
2. Fig 1 also shows the effect of various growth factors on the phosphorylation of FoxO3. However, only data for non-IPF fibroblasts is shown. What is the effect of these growth factors in IPF-fibroblasts? Is there a graded response dependent on the level of FOXO3 phosphorylation typically recorded for different donor lines? Does FoxO3 phosphorylation vary from passage to passage and in response to growth factor stimulation in both IPF and non-IPF fibroblasts?
3. Fig 2C: What is the effect of siRNA knockdown on baseline matrix gene expression? Without this analysis, it is difficult to determine whether this is a specific TGF- β dependent effect or driven by an effect of the absence of FoxO3 on baseline gene expression?

4. Fig 3B. The increase in FoxO3 phosphorylation over time in the bleomycin model is interesting. Is this associated with the conversion to a myofibroblast phenotype over time? It would be very helpful to determine whether ACTA2 expression is higher at 21 versus 14 days in these cells, especially since the authors refer to this as a potential explanation with respect to differences in FoxO3 phosphorylation between saline vs bleomycin-derived fibroblasts.

5. Bleomycin studies: The effect of global and fibroblast-specific FOXO3 knockdown is striking. The histopathology figures presented are however a little worrying. Why is there no saline panel presented for fibroblast-specific FoxO3 knockout mice? More importantly, there appears to be a marked difference in staining between global FoxO3 knockout mice and fibroblast-specific FoxO3 knockout mice (panel G). This requires further exploration. Moreover, given the importance of inflammation as a key driver of the ensuing fibrotic response in the bleomycin model, are there any differences between global versus fibroblast-specific FoxO3 knockout mice in terms of inflammatory response to bleomycin? Further confirmation to support the central conclusion that fibrosis is the key histopathological difference in these bleomycin studies is needed. Assessment of lung hydroxyproline levels is regarded the gold standard endpoint for conclusive assessment of pulmonary fibrosis and should be included as an endpoint in key studies.

6. The authors are commended for attempting to develop a therapeutic approach for specifically inhibiting FoxO3 phosphorylation but regrettably, the UCN-01 data are currently not conclusive. This agent inhibits many phosphokinases in addition to AKT, including calcium-dependent protein kinase C, and cyclin-dependent kinases. It is therefore not surprising that it inhibits growth factor-induced fibroblast proliferation. Moreover, Fig 4A shows that there is no concentration-response for this agent in the 5% FCS proliferation assay. Detailed concentration response experiments are needed to substantiate the conclusion that this agent predominantly exerts its anti-proliferative and anti-fibrotic effects by selectively inhibiting FoxO3 phosphorylation downstream of PDK1/PI3K/AKT. The use of wortmanin at a single concentration of 500nM should also be revisited to confirm the importance of PI3K in mediating the pro-fibrotic effects of TGF-beta since this inhibitor is also known to act on other kinases.

7. The effect of UCN-01 in the bleomycin model is striking and suggests that at the higher concentration, fibrosis is completely inhibited. In contrast, the Ashcroft score data suggests that UCN-01 might not be as effective in inhibiting the fibrotic response. Again, confirmation of these key data by assessing lung hydroxyproline levels is need to substantiate the observation that this agent is principally anti-fibrotic in this model. Additionally, data to demonstrate that the anti-fibrotic effect is principally mediated via modulation of pAKT and FoxO3 in this model are needed to support the main conclusion and therapeutic implications proposed in this manuscript.

Minor comments:

1. Could the authors explain on what basis the two concentrations of UCN-01 and dosing strategy were chosen? How do these relate in terms of the clinical trials performed with this agent in the cancer setting? What is known about the in vivo half-life of this agent?
2. The use of the term "reconstitution" of FoxO3 could lead to confusion as this suggests that this transcription factor has been re-expressed by genetic manipulation rather than modulated by pharmacological inhibition by UCN-01.

Referee #3 (Comments on Novelty/Model System):

The experiments are well done, are logical and of high quality. Overall the findings are somewhat incremental. There has been previous data on FoxO3 and lung fibrosis, but the mice data is new. The fibroblast specific KO is known to be leaky and the author's fail to show specificity to fibroblasts and so it not convincing that the effects of FoxO3 are fibroblast specific. UCN-01 inhibits multiple protein kinase C isoenzymes and so there are likely off target effects. Finally there is no mechanism offered how FoxO3 mediates its fibrotic effects.

Referee #3 (Remarks):

General Comment

In the present study, the authors provide evidence for an important role of FoxO3 in the development of pulmonary fibrosis. In a series of well done in vitro, ex vivo and in vivo experiments using global and tissue specific knockout mice, they show that loss of nuclear FoxO3 activity causes phenotypic changes of the fibroblasts with enhanced proliferation and differentiation to myofibroblasts as well as increased severity of pulmonary fibrosis in the mouse bleomycin model. In addition use of UCN-01, a putative pharmacological tool for nuclear FoxO3 reconstitution, blocked the fibroblast phenotypic change in vitro and prevented lung fibrosis in vivo. Thus the authors' conclude that rescuing FoxO3 could be an effective therapeutic strategy for pulmonary fibrosis.

The experiments are technically well done and encompass an impressive breadth of in vitro, ex vivo and in vivo studies. Overall the paper provides convincing evidence for a crucial role of FoxO3 in the pathogenesis of pulmonary fibrosis. Further they suggest that the effect of FoxO3 may be fibroblast specific. In addition experiments with UCN-01 suggest a novel therapeutic strategy with a drug already approved for clinical trials in resistant cancers.

An important role for FoxO3 in lung fibrosis has been previously reported in multiple publications, so the findings are somewhat incremental, though this paper is the first to report its impact in the mouse bleomycin model of lung fibrosis. The data on the fibroblast-specific KO of FoxO3 used a strategy that is notoriously leaky and thus the data is not fully convincing that the impact of FoxO3 in lung fibrosis is fibroblast specific. In addition the excess mortality of the FoxO3 KO, in both global and fibroblast specific mice, is not well discussed or explained and makes interpretation of the increased fibrosis problematic. The data suggesting that UCN-01 mediates its protective effects only thru reconstitution of Fox-O3 is not convincing since it can inhibit multiple isoenzymes of protein kinase C and the data presented does not exclude the possibility of off target effects. This is suggested by the finding that in the presence of FoxO siRNA, there is still a potent effect of UCN-01 on fibroblast proliferation in response to 5% FCS. Lastly there is no data presented on the mechanism by which reduction of FoxO3 facilitates myofibroblast differentiation or lung fibrosis.

Specific Comments

- The paper would be enhanced showing the deletion of FoxO3 was fibroblast specific.
- Does UCN-01 fail to rescue fibrosis in the FoxO3 KO mice? This is an important experiment as it would provide in vivo specificity that UCN-01 mediates its protection via FoxO3.
- The data that UCN-01 rescues nuclear exclusion of FoxO3 is not particularly convincing.

Additional correspondence (author)

27 July 2016

We would like to thank you for your encouraging response to our manuscript EMM-2016-06261; we appreciate the opportunity provided to respond to your concerns as well as to those of the reviewers. We performed and performing a series of additional experiments to carefully address the comments specified by you and the reviewers.

I would like to inform you that due to low breeding potential of FoxO3 global and fibroblast-specific mice, we were not able to perform the animal experiments requested by Reviewer #3. We are waiting enough transgenic mice and will continue our experiments and will be ready with the resubmission in next 3 months time. We are extremely unhappy with the situation, but were helpless.

I hope you can understand us and provide additional time to resubmit our work.

1st Revision - authors' response

08 August 2017

Referee #1

Major comments:

1. Fig 1D and 1E are critical in that they set the premise for all the ensuing work addressing the importance of FoxO3 as a key regulator of fibroblast responses in IPF. Regrettably, the data presented are not particularly convincing in terms of reporting key differences between the levels of FoxO3 expression in IPF versus non-IPF fibroblast lines (Fig 1D) in that the phosphorylation state of FoxO3 appears to be highly heterogeneous in the IPF lines with only 2/4 showing high levels of phosphorylation. This reviewer is therefore not convinced that these data support the conclusion that this is unique to IPF and would argue that fibroblast heterogeneity might provide explain the reported differences across donor and IPF lines. Confirmatory studies of FoxO3 expression and phosphorylation status based on detailed analysis of patient biopsy tissue are absolutely essential to support the current proposition that this is an important disease-relevant and targetable pathway.

R1: We thank the reviewer for the comment. We would like to mention that the blots provided in the initial manuscript are representative blots while the quantification was carried out from a higher number (6-7 in each group). Hence, the quantification of all the samples as provided in the **Fig. 1F** clearly shows an overall increased phosphorylation of FoxO3 in IPF fibroblasts as compared with donors. However, to further support this data, we carried out screening from more IPF and donor fibroblasts and the data is provided in the **Fig. 1** of the revised manuscript. The blot clearly shows a strong increase in FoxO3 phosphorylation (Thr32) in IPF fibroblasts (IPF-HLF) compared to donors (N-HLF), thus confirming our findings. The result is added to **Fig. 1** in revised manuscript. The variations in FoxO3 phosphorylation among IPF fibroblasts may explain by local concentration of cytokines and growth factor milieu they exposed *in vivo*. However, patient biopsy tissue collection, isolation of fibroblasts from these tissues and the analysis of FoxO3 expression are beyond the scope this work.

Fig. 1. Expression of p-FoxO3 in N-HLF and IPF-HLF (A) Western blot of p-FoxO3 (Thr32) and GAPDH (B) Densitometry quantified data of p-FoxO3 (Thr32) to GAPDH expression ratio (n=4-5/group). Data were analyzed using Student's t-test, * $p < 0.05$, ** $p < 0.01$, *** $p < 0.0001$ versus donor.

2. Fig 1 also shows the effect of various growth factors on the phosphorylation of FoxO3. However, only data for non-IPF fibroblasts is shown.

2.1 What is the effect of these growth factors in IPF-fibroblasts?

R2.1. As suggested by the reviewer, we stimulated IPF fibroblasts (IPF-HLF; n =3) with various growth factors to study the effect on FoxO3 phosphorylation. We observed a strong and similar increase in FoxO3 phosphorylation on stimulation with all the stimuli (5% FCS, PDGF-BB and IGF-1), similar to the effect observed in non-IPF fibroblasts (**Fig. 1G-I**). The result is provided in **Fig. EV 5** of the revised version.

Fig. EV5. (A) Representative western blots of p-FoxO3 (Thr32), FoxO3 and GAPDH in serum starved (48hrs) IPF-HLF (n=3) that were stimulated with 5%FCS or PDGF-BB or IGF-1 as indicated for 30min. Densitometry quantified data of p-FoxO3 (Thr32) to FoxO3 expression ratios, represented as a fold change to non-stimulated cells. Data were analyzed using repeated-measures one-way ANOVA * $p < 0.05$, ** $p < 0.01$, *** $p < 0.001$ versus vehicle treated cells.

2.2 Is there a graded response dependent on the level of FOXO3 phosphorylation typically recorded for different donor lines?

R2.2: In order to answer this question, we stimulated three different non-IPF fibroblasts (N-HLF) with various stimuli (5% FCS, PDGF-BB and IGF-1) and checked for FoxO3 phosphorylation by western blotting. The results are included in **Fig. EV6** of the revised manuscript. The data provided indicates that all three-donor fibroblasts show an increased FoxO3 phosphorylation in response to various growth stimuli. There were no significant differences observed in response of different donor lines to various stimuli in terms of FoxO3 phosphorylation.

Fig. EV6. Western blots of p-FoxO3 (Thr32), FoxO3 and GAPDH in serum starved (48hrs) N-HLF (n=3) that were stimulated with 5%FCS (A) or PDGF-BB (B) or IGF-1 (C) as indicated. Densitometry quantified data of p-FoxO3 (Thr32) to FoxO3 expression ratios, represented as a fold change to non-stimulated cells.

2.3 Does FoxO3 phosphorylation varies from passage to passage and in response to growth factor stimulation in both IPF and non-IPF fibroblasts?

R2.3: We appreciate the comment from the reviewer. To investigate passage dependent differences in FoxO3 phosphorylation by growth factor stimulation, N-HLF and IPF-HLF were stimulated with various growth factors (5% FCS, PDGF-BB and IGF-1) at three different passages (passage 4 to 6). The result is provided in **Fig. EV7** of the revised version. Overall, all the stimulations led to induction of FoxO3 phosphorylation and there was no passage dependent change in response to growth factor stimulation observed in N-HLF and also in IPF-HLF. This finding therefore proves that the response to growth factor stimulation is considerably stable over passaging of the cells.

Fig. EV7. Western blots of p-FoxO3 (Thr32), FoxO3 and GAPDH in serum starved (48hrs) (A) N-HLF or (B) IPF-HLF that were stimulated with 5%FCS or PDGF-BB or IGF-1 over different passages (P3 to P5) as indicated.

Furthermore, N-HLF and IPF-HLF (n= 3 each) were stimulated with different growth stimuli (5% FCS, PDGF-BB and IGF-1) to study if there is difference in response to various growth factors in terms of FoxO3 phosphorylation. The representative blots from IPF-HIF are provided in **Fig. EV5** while the result from the N-HLF is provided in the **Fig. EV8** of the revised manuscript. We clearly observe a strong increase in FoxO3 phosphorylation by all three stimuli to a similar extent in both N-HLF and IPF-HLF both.

Fig. EV8. Representative western blots of p-FoxO3 (Thr32), FoxO3 and GAPDH in serum starved (48hrs) N-HLF (n=3) that were stimulated with 5%FCS or PDGF-BB or IGF-1 as indicated for 30min. Densitometry quantified data of p-FoxO3 (Thr32) to FoxO3 expression ratios, represented as a fold change to non-stimulated cells. Data were analyzed using repeated-measures one-way ANOVA * $p < 0.05$, ** $p < 0.01$, *** $p < 0.001$ versus vehicle treated cells.

3. Fig 2C: What is the effect of siRNA knockdown on baseline matrix gene expression? Without this analysis, it is difficult to determine whether this is a specific TGF- β dependent effect or driven by an effect of the absence of FoxO3 on baseline gene expression?

R3: Indeed, the effect of siRNA knockdown on matrix gene expression (**Fig. 2C**) is studied at baseline level, without any TGF- β stimulation. Hence, FoxO3 depletion itself is sufficient to direct the donor fibroblasts towards a pro-fibrotic phenotype.

Further, to explore the mechanism involved in FoxO3 mediated matrix gene expression; we first analyzed their promoter sites (Col1A1, Col3A1) for presence of FoxO3 binding sites. However, we were unable to find any binding sites, hinting towards an indirect regulation of these genes by FoxO3. FoxO3 is shown to control expression of various transcription and co-transcription factors involved in regulation of differentiation. Interestingly, FoxO3 is shown to directly bind to myocardin (MYOCD) promoter and inhibit its expression in human aortic smooth muscle cells, preventing smooth muscle cell differentiation (*Yang et al., 2013, PLoS ONE 8(3): e58746*). Myocardin family members interact with Serum response factor (SRF) as homo- or hetero- dimers and stimulate transcription via conserved CArG box DNA elements. In a study focussing on liver fibrosis, myocardin was shown to increase expression of myofibroblast related markers including SMA and type I collagen. It was further upregulated during in rat primary hepatic stellate cells during *in vitro* activation and in fibrotic liver of a dimethylnitrosamine -induced fibrosis rat model, demonstrating regulatory role of myocardin during hepatic stellate cells activation and the pathogenesis of liver fibrosis (*Shimada et al., Liver International, 2010; Vol 20, Iss. 1:42-54*). This prompted us to examine whether in a similar fashion, FoxO3 negatively regulates myocardin that in turn controls fibroblast-myofibroblast differentiation. We observed a strong increase in expression of myocardin in N-HLFs with FoxO3 knockdown, indicating role of myocardin in mediating effect of FoxO3 in regulating myofibroblast differentiation in pulmonary fibrosis. The result is provided as **Fig. EV12** in revised manuscript.

Fig. EV12. N-HLFs (n=3) were transfected with scramble siRNA or FoxO3 siRNA. mRNA expression of MYOCD was analyzed by qPCR. Data were analyzed using Student's t test, * $p < 0.05$, ** $p < 0.01$, *** $p < 0.001$ versus scramble siRNA group.

4. Fig 3B. The increase in FoxO3 phosphorylation over time in the bleomycin model is interesting. Is this associated with the conversion to a myofibroblast phenotype over time? It would be very helpful to determine whether ACTA2 expression is higher at 21 versus 14 days in these cells, especially since the authors refer to this as a potential explanation with respect to differences in FoxO3 phosphorylation between saline vs bleomycin-derived fibroblasts.

R4: We thank the reviewer for the comment and as suggested, we carried out western blotting for ACTA2 (a-Sma) to determine its expression in mouse fibroblasts isolated from saline and bleomycin treated (14 days and 21 days) mice. The data is added to **Fig. 3B** of the revised manuscript. We observed a strong increase a-Sma expression in fibroblasts isolated from bleomycin treated mice as compared to saline. This finding directly co-relates with increased FoxO3 phosphorylation observed in bleomycin treated mice fibroblasts indicating that increase in a-Sma is an outcome of decreased FoxO3 expression.

Fig. 3B. Fibroblasts were isolated from saline-treated mice lungs at day 21 post-instillation and mice lungs after 14 or 21 days of bleomycin instillation. Representative western blot of α -Sma protein level. Actb was used as a loading control. Densitometry quantified data of α -Sma to Actb expression ratio (n=2/group). Data were analyzed using repeated-measures one-way ANOVA * $p < 0.05$, ** $p < 0.01$, *** $p < 0.001$ versus Saline treated group.

5. Bleomycin studies: The effect of global and fibroblast-specific FOXO3 knockdown is striking.

5.1. The histopathology figures presented are however a little worrying. Why is there no saline panel presented for fibroblast-specific FoxO3 knockout mice? More importantly, there appears to be a marked difference in staining between global FoxO3 knockout mice and fibroblast-specific FoxO3 knockout mice (panel G). This requires further exploration.

R5.1: The saline panel is now included in the revised figure (**Fig. 3** of the revised version). Similar to *Foxo3*^{-/-} mice, *Foxo3_{f.b}*^{-/-} did not exhibit any overt phenotype and no body weight changes similar to that of wild type littermates. Re-performing the lung function measurement in WT, *Foxo3*^{-/-} and

Foxo3_{fb}^{-/-} mice that were either instilled with saline and bleomycin confirms no lung functional parameters and fibrotic changes in saline instilled mice. However, an aggravated fibrosis phenotype was observed in *Foxo3*^{-/-} and *Foxo3_{fb}*^{-/-} mice compared to WT bleomycin-instilled mice.

As pointed out by the reviewer, marked difference in staining between global FoxO3 knockout mice and fibroblast-specific FoxO3 knockout mice is due to performing of the stainings at different time point. We re-performed the H&E stainings from all the groups at the same time and Figure is revised accordingly (**Fig. 3H**). In accordance, *Foxo3*^{-/-} and *Foxo3_{fb}*^{-/-} bleomycin-instilled mouse lungs demonstrated a significant increase in hydroxyproline content and fibrotic score as compared to WT littermates (**Fig. 3G-H**).

5.2: Moreover, given the importance of inflammation as a key driver of the ensuing fibrotic response in the bleomycin model, are there any differences between global versus fibroblast-specific FoxO3 knockout mice in terms of inflammatory response to bleomycin?

R5.2: We agree with the reviewer that the inflammation is a key driver of the ensuing fibrotic response in the bleomycin model. To study the inflammatory responses in wild type, global (*Foxo3*^{-/-}) versus fibroblast-specific FoxO3 (*Foxo3_{fb}*^{-/-}) knockout mice, we performed immunohistochemical staining for CD68 (macrophages), CD3 (T cells) and CD45 (leukocytes). We observed that a marked infiltrate of CD68-, CD3- and CD45- positive cells, localized to sites of injury and fibrosis within the lungs of bleomycin-challenged WT, *Foxo3*^{-/-} and *Foxo3_{fb}*^{-/-} mouse lungs (**Fig. EV14A-C**). Interestingly, an augmented increase in CD3- positive cells was found in both *Foxo3*^{-/-} and *Foxo3_{fb}*^{-/-} mouse lungs compared to WT mouse lungs (**Fig. EV14D**).

Infiltration of T cells in global and fibroblast-specific FoxO3 knockout mice is intriguing and may suggest that FoxO3 depletion in fibroblasts may promote recruitment of T cells to the fibrotic area by secreting chemokines and promote chronic inflammation. Further studies are needed to clarify this underlying mechanism.

Fig. EV14. Immunofluorescence staining was performed on WT, *Foxo3^{-/-}* and *Foxo3_{fb}^{-/-}* mice lung sections (saline and bleomycin instilled) using CD68, CD45 and CD3 antibodies. Representative pictographs depicting CD68 (A), CD45 (B) and CD3 (C) staining in green from mice (n=3) in each group. DAPI was used as a nuclear stain. Scale = 50µm. Fluorescence intensities of CD3 stained

sections (n=5/6 per group) were quantified using ImageJ software and normalized to DAPI intensity. Data were analyzed using repeated-measures one-way ANOVA *p<0.05, **p<0.01, ***p<0.001 versus WT saline group and §p<0.05, §§p<0.01, §§§p<0.001 versus WT bleomycin group. m1, m2 and m3 represents 3 different mice evaluated in each group.

5.3: Further confirmation to support the central conclusion that fibrosis is the key histopathological difference in these bleomycin studies is needed. Assessment of lung hydroxyproline levels is regarded the gold standard endpoint for conclusive assessment of pulmonary fibrosis and should be included as an endpoint in key studies.

R5.3: As suggested by the reviewer, we measured the lung hydroxyproline levels in the following experiments:

(a) In wild type (WT), global (*Foxo3^{-/-}*) and fibroblast-specific FoxO3 (*Foxo3_{fb}^{-/-}*) knockout mice that were instilled with saline or bleomycin.

(b) UCN-01 treated and vehicle-treated WT and *Foxo3^{-/-}* bleomycin instilled mice.

Most importantly, lung hydroxyproline measurements confirmed histopathological differences in these mice. *Foxo3^{-/-}* and *Foxo3_{fb}^{-/-}* bleomycin-instilled mouse lungs demonstrated a significant increase in hydroxyproline levels as compared to WT littermates (**Fig. 3G**). On the other hand, a significant decrease of hydroxyproline levels was observed in UCN-01-treated mice as compared to vehicle-treated WT mice; where as this effect of UCN-01 is compromised in *Foxo3^{-/-}* mice (**Fig. 9D**).

6.The authors are commended for attempting to develop a therapeutic approach for specifically inhibiting FoxO3 phosphorylation but regrettably, the UCN-01 data are currently not conclusive. This agent inhibits many phosphokinases in addition to AKT, including calcium-dependent protein kinase C, and cyclin-dependent kinases. It is therefore not surprising that it inhibits growth factor-induced fibroblast proliferation.

6.1 Moreover, Fig 4A shows that there is no concentration-response for this agent in the 5% FCS proliferation assay. Detailed concentration response experiments are needed to substantiate the conclusion that this agent predominantly exerts its anti-proliferative and anti-fibrotic effects by selectively inhibiting FoxO3 phosphorylation downstream of PDK1/PI3K/AKT.

R6.1: We thank the reviewer for the comment. Firstly, the concentrations of UCN-01 used for *the in vitro* studies were based upon the MTT assay carried out in N-HLFs where the IC₅₀ was found to be more than 200nM (**Fig. EV15**). Therefore, following concentrations were used for proliferation assays: 10nM, 50nM, 100nM and 200nM.

Fig. EV15. Human lung fibroblasts from controls (N-HLF) were serum starved for 48 hrs. Then, cells were treated with UCN-01 (10, 50, 100 and 200nM) or vehicle (DMSO) or left untreated and cell viability was assessed by MTT assay after 24 hrs. Data are represented as percentage of control; vehicle treated cells. Bars indicate means± S.E.M (n=3). Red line indicates IC₅₀.

These concentrations showed a dose dependent decrease in proliferation of IPF-HLFs (**Fig. 8A** of the revised version). On the other hand, we also observe a strong decrease in proliferation of N-HLFs treated with UCN-01 but without dose dependency. This could be due to the fact that IPF-HLFs are more proliferative than N-HLFs. Hence; lower concentrations of the inhibitor are sufficient to reduce the proliferation of N-HLFs as compared to the IPF-HLFs.

Secondly, as rightly stated by the reviewer, UCN-01 inhibits many kinases in addition to AKT. Interestingly, many studies also revealed that the anti neoplastic activities of UCN-01 could not be explained and do not correlate with its ability to block protein kinase activity (*Patel, V., et al., Clin Cancer Res, 2002. 8(11): p. 3549-60; Jia, W., et al., Blood, 2003. 102(5): p.1824-32*). Further, UCN-01 was shown to inhibit PDK1, an important kinase of PI3K/AKT pathway. After PI3K is activated, it generates phosphatidylinositol 3,4,5-triphosphate (PtdIns(3,4,5)P₃). In turn (PtdIns(3,4,5)P₃) recruits AKT to the plasma membrane where AKT is phosphorylated (activated) on two key regulatory sites: at Ser473 by mTOR (*Sarbassov, D.D., et al., Science, 2005. 307(5712): p. 1098-101*) and at Thr308 by PDK1 (*Stephens, L., et al., Science, 1998. 279(5351): p.710-4*). Phosphorylation at both sites is necessary for full activation of AKT and its subsequent effects. We were able to show that UCN-01 inhibits the AKT phosphorylation at Thr308 under various growth factor stimulation i.e. 5% FCS, IGF-1, PDGF-BB (**Fig. 5** of the revised manuscript). In order to analyze how much anti-proliferative and anti-fibrotic effect of UCN-01 is mediated by selectively inhibiting FoxO3 phosphorylation downstream of PDK1/PI3K/AKT, we overexpressed constitutively active AKT deletion mutant (PH domain deletion mimics constant phosphorylation, hence, constant activation) in combination with UCN-01 under 5% FCS and TGFβ stimulation. We observed that constitutive activation of AKT reversed the decrease observed in FoxO3 phosphorylation induced by UCN-01 under both 5% FCS (**Fig. S20A**) and TGFβ (**Fig. 9E**) to a substantial extent. This clearly indicates that UCN-01 decreases FoxO3 phosphorylation by inhibiting PDK1/AKT pathway. Additionally, constitutive active AKT was able to reverse the anti-proliferative effect of UCN-01 under 5% FCS stimulation substantially (**Fig. S20B**). Interestingly, it was also able to significantly increase the expression of fibrotic markers (Col1A1, Col3A1) that was decreased by UCN-01 treatment with TGFβ stimulation (**Fig. 9F**). These findings undoubtedly signify that majority of effect of UCN-01 proliferation and fibrosis is mediated via PDK1/AKT controlled FoxO3 phosphorylation.

Fig. EV20. N-HLF were transfected with empty vector (EV) or AKT mutant (AKT mut) plasmid. (A and B) 6 hours after transfection, cells were serum starved for 36 hrs and then stimulated with 5%FCS in presence or absence of UCN-01. From the above-treated samples, after 24 hrs western blots (p-FoxO3 (Thr32), FoxO3, AKT, AKT mut, GAPDH) and cell proliferation measurements (BrdU incorporation) were performed. Data represent percentage of control, EV non-stimulated cells (n=3). Data were analyzed using one-way ANOVA, ***p<0.05, and **p<0.01, and *p<0.001 versus 5%FCS-EV and §p<0.05, §§p<0.01, §§§p<0.001, versus UCN-01-EV.

6.2 The use of wortmanin at a single concentration of 500nM should also be revisited to confirm the importance of PI3K in mediating the pro-fibrotic effects of TGF-beta since this inhibitor is also known to act on other kinases.

R6.2: We thank the reviewer for the comment. To address this comment, we treated the N-HLFs (n=3) with two concentrations of wortmannin (250nM and 500nM) in presence of TGF β . TGF β stimulation led to a significant increase in AKT phosphorylation and subsequent FoxO3 phosphorylation. Wortmannin treatment led to a strong inhibition of TGF β induced AKT phosphorylation at Threonine 308, site specifically phosphorylated by PDK1 (*Stephens, L., et al., Science, 1998. 279(5351): p.710-4*). Further, we observed a concomitant reduction in TGF β induced FoxO3 phosphorylation by both concentrations in a dose dependent manner (**Fig. EV18**). These results indicate that wortmannin inhibits PDK1 mediated AKT phosphorylation under TGF β stimulation leading to inhibition of FoxO3 phosphorylation.

Though we agree with the reviewer that wortmannin does have documented inhibitory effects on other kinases. For our studies, we have used the concentrations of wortmannin already described in many publications for *in vitro* cell based AKT inhibition (*Wang, Q., JBC, 2002. 277, 36602-36610*).

Fig. EV18. Representative western blots of p-FoxO3 (Thr32), FoxO3, p-AKT (Thr308), AKT and GAPDH in serum starved (48hrs) IPF-HLF (n=3) that were stimulated with TGF β -1 for 4hrs in medium containing Wortmanin (Wort.) (250nM or 500nM) or UCN-01 (50nM) as indicated. Densitometry quantified data of p-FoxO3 (Thr32) to FoxO3 and p-AKT (Thr308) to AKT expression ratios, represented as a fold change to non-stimulated cells. Data were analyzed using repeated-measures one-way ANOVA *p<0.05, **p<0.01, ***p<0.001 versus TGF β treated cells.

7. The effect of UCN-01 in the bleomycin model is striking and suggests that at the higher concentration, fibrosis is completely inhibited. In contrast, the Ashcroft score data suggests that UCN-01 might not be as effective in inhibiting the fibrotic response. Again, confirmation of these key data by assessing lung hydroxyproline levels is need to substantiate the observation that this agent is principally anti-fibrotic in this model. Additionally, data to demonstrate that the anti-fibrotic effect is principally mediated via modulation of pAKT and FoxO3 in this model are needed to support the main conclusion and therapeutic implications proposed in this manuscript.

R7: As suggested by the reviewer, we measured the lung hydroxyproline levels in the UCN-01 treated and vehicle-treated bleomycin instilled mice. Consistent with the histopathology and lung functions measurements, a significant decrease of hydroxyproline levels was observed in UCN-01-treated mice as compared to vehicle-treated mice (**Fig. 9D**). Additionally to demonstrate the anti-fibrotic effects are principally mediated via FoxO3, FoxO3 knockout (*Foxo3^{-/-}*) mice were treated with UCN-01. Importantly, FoxO3 depletion diminished the anti-fibrotic effects of UCN-01, suggesting that UCN-01 mediates majorly its anti-fibrotic effects via FoxO3 *in vivo* (**Fig. 9D**).

Further, as describe in **R 6.1** in order to analyze how much anti-fibrotic effect of UCN-01 is mediated via modulation of pAKT and FoxO3 *in vitro*, we overexpressed constitutively active AKT deletion mutant (PH domain deletion mimics constant phosphorylation, hence, constant activation) in combination with UCN-01 under TGF β stimulation. We observed that constitutive activation of AKT reversed the decrease observed in FoxO3 phosphorylation induced by UCN-01 under TGF β to

a substantial extent (**Fig. 9E**). This clearly indicates that UCN-01 decreases FoxO3 phosphorylation by inhibiting PDK1/AKT pathway. Interestingly, overexpression of constitutively active AKT deletion mutant was also able to significantly increase the expression of fibrotic markers (CollA1, Col3A1) that was decreased by UCN-01 treatment with TGF β stimulation (**Fig. 9D**). These findings undoubtedly signify that majority of effect of UCN-01 fibrosis is mediated via PDK1/AKT controlled FoxO3 phosphorylation.

Minor comments:

8. Could the authors explain on what basis the two concentrations of UCN-01 and dosing strategy were chosen? How do these relate in terms of the clinical trials performed with this agent in the cancer setting? What is known about the *in vivo* half-life of this agent?

R8: The dosing of the UCN-01 in preclinical studies is chosen based on the following publications:

<https://www.ncbi.nlm.nih.gov/pubmed/12196722>

<https://www.ncbi.nlm.nih.gov/pubmed/?term=21036719>

<https://www.ncbi.nlm.nih.gov/pubmed/?term=24810059>

The dosing used in clinical trials and pharmacokinetics data are shown in the following publication:

<https://www.ncbi.nlm.nih.gov/pmc/articles/PMC3557498/>

This study suggests that UCN-01 exhibited a long half-life (292 h).

9. The use of the term "reconstitution" of FoxO3 could lead to confusion as this suggests that this transcription factor has been re-expressed by genetic manipulation rather than modulated by pharmacological inhibition by UCN-01.

R9: As suggested by the reviewer, the title has been revised as "FoxO3 an important player in fibrogenesis and therapeutic target for idiopathic pulmonary fibrosis"

Referee #3 (Comments on Novelty/Model System):

General Comment

1. An important role for FoxO3 in lung fibrosis has been previously reported in multiple publications, so the findings are somewhat incremental, though this paper is the first to report its impact in the mouse bleomycin model of lung fibrosis.

R1: We would like to thank for all the valuable comments. As pointed out, this is the first report studying the impact of FoxO3 *in vivo* in the mouse bleomycin model of lung fibrosis employing global (*Foxo3^{-/-}*) and fibroblast-specific (*Foxo3_{fb}^{-/-}*) FoxO3 knockout mice. In addition, this study provides a new means of activating FoxO3 *in vivo* by UCN-01 to attenuate the IPF myofibroblast phenotype *in vitro* and bleomycin induced lung fibrosis *in vivo*. Most importantly, additional experiments performed for revision work suggest that UCN-01 mediates majorly its protection via FoxO3 *in vivo*.

1.1 The data on the fibroblast-specific KO of FoxO3 used a strategy that is notoriously leaky and thus the data is not fully convincing that the impact of FoxO3 in lung fibrosis is fibroblast specific.

R1.1: To show that the deletion of FoxO3 is fibroblast specific, lung fibroblasts and type II alveolar cells were isolated from the fibroblast specific Foxo3 knockout (*Foxo3_{fb}^{-/-}*) and wildtype (WT) littermates. Foxo3 expression was analyzed by qPCR after RNA isolation. There was a significant downregulation of Foxo3 observed in fibroblasts but not in alveolar type II cells isolated from *Foxo3_{fb}^{-/-}* mice (**Fig. EV13A-B**).

In addition, Foxo3 immunohistochemical staining displayed low Foxo3 expression in fibroblasts, but not in any other lung parenchymal cells in *Foxo3_{fb}^{-/-}* mouse lung sections compared WT mouse lung sections (**Fig EV13C**), proving the fibroblast specificity of the knockout in both saline and

bleomycin instilled mice.

Fig. EV13. Fibroblasts and alveolar type II epithelial cells were treated were isolated from *Foxo3fb*^{-/-} and WT littermates. mRNA expression analysis of Foxo3 by qPCR (n=3-4/5 per group). Data were analyzed using Student's t test, *p<0.05, **p<0.01, ***p<0.001 versus WT group. Immunofluorescence staining was performed on WT and *Foxo3fb*^{-/-} mice lung sections (saline and bleomycin instilled) using FoxO3 and α-SMA antibodies. (C) Representative pictographs showing FoxO3 staining in green, with α-SMA stained in red. DAPI was used as nuclear stain. Scale = 50μm.

1.2 In addition the excess mortality of the FoxO3 KO, in both global and fibroblast specific mice, is not well discussed or explained and makes interpretation of the increased fibrosis problematic.

R1.2: On the other hand, *Foxo3*^{-/-}, *Foxo3fb*^{-/-} bleomycin-instilled mice showed a decrease in survival at day 12 of post-bleomycin instillation, whereas no death in WT mice was observed (**Fig. 3C**). This mortality of *Foxo3*^{-/-}, *Foxo3fb*^{-/-} bleomycin-instilled mice can be explained by severe fibrosis observed in the lungs that were harvested after the death of these mice (data not shown). However, despite mortality, for the lung function measurements in *Foxo3* mutant-bleomycin treated mice, the experiments were terminated at day 14. This time point was chosen to study the role of FoxO3 in the fibrotic phase and to exclude the inflammatory phase of bleomycin-induced lung fibrosis. These results indicate that FoxO3 has an important role in attenuating the severity of lung fibrosis induced by bleomycin instillation.

1.3 The data suggesting that UCN-01 mediates its protective effects only through reconstitution of Fox-O3 is not convincing since it can inhibit multiple isoenzymes of protein kinase C and the data presented does not exclude the possibility of off target effects. This is suggested by the finding that in the presence of FoxO siRNA, there is still a potent effect of UCN-01 on fibroblast proliferation in response to 5% FCS.

R1.3: As rightly stated by the reviewer, UCN-01 inhibits many kinases in addition to AKT. Interestingly, many studies also revealed that the anti neoplastic activities of UCN-01 could not be explained and do not correlate with its ability to block protein kinase activity (*Patel, V., et al., Clin*

Cancer Res, 2002. 8(11): p. 3549-60; Jia, W., et al., *Blood*, 2003. 102(5): p.1824-32). Further, UCN-01 was shown to inhibit PDK1, an important kinase of PI3K/AKT pathway. After PI3K is activated, it generates phosphatidylinositol 3,4,5-triphosphate (PtdIns(3,4,5)P₃). In turn (PtdIns(3,4,5)P₃) recruits AKT to the plasma membrane where AKT is phosphorylated (activated) on two key regulatory sites: at Ser473 by mTOR (Sarbasov, D.D., et al., *Science*, 2005. 307(5712): p. 1098-101) and at Thr308 by PDK1 (Stephens, L., et al., *Science*, 1998. 279(5351): p.710-4). Phosphorylation at both sites is necessary for full activation of AKT and its subsequent effects. We were able to show that UCN-01 inhibits the AKT phosphorylation at Thr308 under various growth factor stimulation i.e. 5% FCS, IGF-1, PDGF-BB (Fig. 5 of the revised manuscript). In order to analyze how much anti-proliferative and anti-fibrotic effect of UCN-01 is mediated by selectively inhibiting FoxO3 phosphorylation downstream of PDK1/PI3K/AKT, we overexpressed constitutively active AKT deletion mutant (PH domain deletion mimics constant phosphorylation, hence, constant activation) in combination with UCN-01 under 5% FCS and TGFβ stimulation. We observed that constitutive activation of AKT reversed the decrease observed in FoxO3 phosphorylation induced by UCN-01 under both 5% FCS (Fig. EV20A) and TGFβ (Fig. 9E) to a substantial extent. This clearly indicates that UCN-01 decreases FoxO3 phosphorylation by inhibiting PDK1/AKT pathway. Additionally, constitutive active AKT was able to reverse the anti-proliferative effect of UCN-01 under 5% FCS stimulation substantially (Fig. EV20B). Interestingly, it was also able to significantly increase the expression of fibrotic markers (Col1A1, Col3A1) that was decreased by UCN-01 treatment with TGFβ stimulation (Fig. 9F). These findings undoubtedly signify that majority of effect of UCN-01 proliferation and fibrosis is mediated via PDK1/AKT controlled FoxO3 phosphorylation.

Fig. EV20. N-HLF were transfected with empty vector (EV) or AKT mutant (AKT mut) plasmid. (A and B) 6 hours after transfection, cells were serum starved for 36 hrs and then stimulated with 5%FCS in presence or absence of UCN-01. From the above-treated samples, after 24 hrs western blots (p-FoxO3 (Thr32), FoxO3, AKT, AKT mut, GAPDH) and cell proliferation measurements (BrdU incorporation) were performed. Data represent percentage of control, EV non-stimulated cells (n=3). Data were analyzed using one-way ANOVA, ***p<0.05, and **p<0.01, and *p<0.001 versus 5%FCS-EV and §p<0.05, §§p<0.01, §§§p<0.001, versus UCN-01-EV.

1.4 Lastly there is no data presented on the mechanism by which reduction of FoxO3 facilitates myofibroblast differentiation or lung fibrosis.

R1.4: We thank the reviewer for the question and in order to answer it, we first analyzed the promoter sites of various fibroblast-myofibroblast differentiation markers (Col1A1, Col3A1) for presence of FoxO3 binding sites. However, we were unable to find any binding sites, hinting towards an indirect regulation of these genes by FoxO3. FoxO3 is shown to control expression of various transcription and co-transcription factors involved in regulation of differentiation. Interestingly, FoxO3 is shown to directly bind to myocardin (MYOCD) promoter and inhibit its expression in human aortic smooth muscle cells, preventing smooth muscle cell differentiation (Yang et al., 2013, *PLoS ONE* 8(3): e58746). Myocardin family members interact with Serum

response factor (SRF) as homo- or heterodimers and stimulate transcription via conserved CARG box DNA elements. In a study focussing on liver fibrosis, myocardin was shown to increase expression of myofibroblast related markers including SMA and type I collagen. It was further upregulated during in rat primary hepatic stellate cells during *in vitro* activation and in fibrotic liver of a dimethylnitrosamine-induced fibrosis rat model, demonstrating regulatory role of myocardin during hepatic stellate cells activation and the pathogenesis of liver fibrosis (*Shimada et al., Liver International, 2010; Vol 20, Iss. 1:42-54*). This prompted us to investigate whether in a similar fashion, FoxO3 negatively regulates myocardin that in turn controls fibroblast-myofibroblast differentiation. We observed a strong increase in expression of myocardin in N-HLFs with FoxO3 knockdown, indicating role of myocardin in mediating effect of FoxO3 in regulating myofibroblast differentiation in pulmonary fibrosis. The result is provided as **Fig. EV12** in revised manuscript.

Fig. EV12. N-HLFs (n=3) were transfected with scramble siRNA or FoxO3 siRNA. mRNA expression of MYOCD was analyzed by qPCR. Data were analyzed using Student's t test, *p<0.05, **p<0.01, ***p<0.001 versus scramble siRNA group.

Specific Comments

2. The paper would be enhanced showing the deletion of FoxO3 was fibroblast specific.

R.2: To show that the deletion of FoxO3 is fibroblast specific, lung fibroblasts and type II alveolar cells were isolated from the fibroblast specific Foxo3 knockout (*Foxo3_{fb}^{-/-}*) and wildtype (WT) littermates. Foxo3a expression was analyzed by qPCR after RNA isolation. There was a significant downregulation of Foxo3 observed in fibroblasts but not in alveolar type II cells isolated from *Foxo3_{fb}^{-/-}* mice (**Fig. EV13A-B**).

In addition, Foxo3 immunohistochemical staining displayed low Foxo3 expression in fibroblasts, but not in any other lung parenchymal cells in *Foxo3_{fb}^{-/-}* mouse lung sections compared WT mouse lung sections (**Fig EV13C**), proving the fibroblast specificity of the knockout in both saline and bleomycin instilled mice.

Fig. EV13. Fibroblasts and alveolar type II epithelial cells were treated were isolated from *Foxo3_{fb}^{-/-}* and WT littermates. mRNA expression analysis of Foxo3 by qPCR (n=3-4/5 per group). Data were analyzed using Student's t test, *p<0.05, **p<0.01, ***p<0.001 versus WT group. Immunofluorescence staining was performed on WT and *Foxo3_{fb}^{-/-}* mice lung sections (saline and bleomycin instilled) using FoxO3 and α-SMA antibodies. (C) Representative pictographs showing FoxO3 staining in green, with α-SMA stained in red. DAPI was used as nuclear stain. Scale = 50μm.

3. Does UCN-01 fail to rescue fibrosis in the FoxO3 KO mice? This is an important experiment, as it would provide *in vivo* specificity that UCN-01 mediates its protection via FoxO3.

R.3: To investigate whether UCN-01 fail to rescue fibrosis in the FoxO3 KO mice, WT and FoxO3 knockout (*Foxo3^{-/-}*) mice were treated with UCN-01 (7.5 mg/Kg) or vehicle on every second day starting at day 7 after the bleomycin challenge and analyzed for the therapeutic benefit of UCN-01 at day 21. In WT bleomycin challenged mice, UCN-01 treatment improved total lung capacity; lung compliance and a significant decrease in tissue resistance and hydroxyproline content as compared to vehicle-treated WT mice (**Fig. 9** of the revised manuscript). On the other hand, *Foxo3^{-/-}* mice challenged with bleomycin had significantly impaired lung function compared to WT bleomycin-instilled mice. Importantly, even in these *Foxo3^{-/-}* mice with aggravated bleomycin-induced pulmonary fibrosis, UCN-01 was failed to rescue fibrosis in the bleomycin instilled *Foxo3^{-/-}* mice. FoxO3 depletion decreased the beneficial effects of UCN-01 more than 50% on total lung capacity and lung compliance (**Fig. 9A-B**). Similarly, it also partly failed to rescue lung tissue resistance and fibrosis (**Fig. 9C-D**). These results suggest that UCN-01 mediates majorly its protection via FoxO3 *in vivo*.

4. The data that UCN-01 rescues nuclear exclusion of FoxO3 is not particularly convincing

R.4: We thank the reviewer for the comment and to substantiate the nuclear exclusion shown by immunocytochemistry in submitted manuscript, we carried out cellular fractionation followed by western blotting in presence or absence of UCN-01 in N-HLFs (n=3) stimulated with 5% FCS. The data is presented in **Fig. EV17**. We clearly observe stronger phospho-FoxO3 (Thr32) in cytosolic fraction as compared to nuclear fraction, proving that phosphorylation of FoxO3 leads to its nuclear exclusion. This FoxO3 phosphorylation is increased with 5% FCS stimulation in both fractions (stronger in cytosolic fraction), which is reversed by UCN-01 treatment.

The FoxO3 blot doesn't show any clear changes at 6 hrs stimulation with 5% FCS. This could be due to the fact that the FoxO3 antibody recognizes total FoxO3 (non-phospho and phospho-FoxO3). Hence, instead of normalizing FoxO3 to the loading controls directly, FoxO3 values were first normalized to phospho-FoxO3 (Thr32) levels. This normalization is necessary in order to remove the interference of phospho-FoxO3 in quantifying active FoxO3 levels. With this normalization, we are able to observe decreased nuclear FoxO3 levels on 5%FCS stimulation as compared to serum starvation conditions. This effect was substantially reversed by UCN-01 treatment. Interestingly, on normalization, we observe a strong increase in nuclear FoxO3 as compared to cytosolic FoxO3 with UCN-01 indicating strong inhibitory effect of UCN-01 on nuclear exclusion of FoxO3.

Fig. EV17: N-HLFs (n=3) were serum starved for 48hrs followed by stimulation with 5% FCS for 6hrs in medium containing UCN-01 (50nM) as indicated. Representative western blots of p-FoxO3 (Thr32), FoxO3, LAMIN B1 and TUBA1B from the cytoplasmic (C) and nuclear fractions (N). Densitometry quantified data of FoxO3 to p-FoxO3 (Thr32) and the ratio was normalized to TUBA1B for cytoplasmic fraction and LAMIN B1 for nuclear fraction. Expression ratios are represented as a fold change to cytosolic fraction of 5% FCS treated cells. Data were analyzed using Student's t test, *p<0.05, **p<0.01, ***p<0.001.

2nd Editorial Decision

06 October 2017

Thank you for the submission of your revised manuscript to EMBO Molecular Medicine. We have now received the enclosed reports from the referees that were asked to re-assess it. As you will see the reviewers are now globally supportive and I am pleased to inform you that we will be able to accept your manuscript pending editorial amendments.

***** Reviewer's comments *****

Referee #1 (Comments on Novelty/Model System for Author):

The authors have addressed my major concerns in this revised manuscript

Referee #3 (Remarks for Author):

The authors have adequately responded to my critiques. The only point I would like them to address is to include the methods by which they isolated mouse fibroblasts from type alveolar type 2 cells in the Foxo3 fb ^{-/-} mice for the fibroblast specificity experiment.

Referee #1 (Comments on Novelty/Model System for Author):

The authors have addressed my major concerns in this revised manuscript.

R: We would like to thank the reviewer for constructive comments and valuable suggestions to delineate the concept of FoxO3 as an important player in fibrogenesis and therapeutic target for idiopathic pulmonary fibrosis.

Referee #3 (Remarks for Author):

The authors have adequately responded to my critiques. The only point I would like them to address is to include the methods by which they isolated mouse fibroblasts from type alveolar type 2 cells in the Foxo3 fb^{-/-} mice for the fibroblast specificity experiment.

R: We would like to thank the reviewer for this reminder as we forgot to include these methods in the manuscript. As suggested by the reviewer, we included in methods section of the revised manuscript, the isolation protocol of fibroblasts and alveolar type II cells from WT and Foxo3 fb^{-/-} mice.

Corresponding Author Name: SONI SAVAI PULLAMSETTI

Manuscript Number: EMM-2016-06261